# Causally Aligned Curriculum Learning

**Mingxuan Li** and **Junzhe Zhang** and **Elias Bareinboim**
Causal Artificial Intelligence Lab, Columbia University, USA
`{ml,junzhez,eb}@cs.columbia.edu`

## Abstract

A pervasive challenge in Reinforcement Learning (RL) is the "curse of dimensionality" which is the exponential growth in the state-action space when optimizing a high-dimensional target task. The framework of *curriculum learning* trains the agent in a curriculum composed of a sequence of related and more manageable source tasks. The expectation is that when some optimal decision rules are shared across source tasks and the target task, the agent could more quickly pick up the necessary skills to behave optimally in the environment, thus accelerating the learning process. However, this critical assumption of invariant optimal decision rules does not necessarily hold in many practical applications, specifically when the underlying environment contains *unobserved confounders*. This paper studies the problem of curriculum RL through causal lenses. We derive a sufficient graphical condition characterizing causally aligned source tasks, i.e., the invariance of optimal decision rules holds. We further develop an efficient algorithm to generate a causally aligned curriculum, provided with qualitative causal knowledge of the target task. Finally, we validate our proposed methodology through experiments in discrete and continuous confounded tasks with pixel observations.

## 1 Introduction

As Roma was not built in one day, learning to achieve a complex task (e.g., cooking, driving) directly can be challenging. Instead, the human learning process is scaffolded with incremental difficulty to support acquiring progressively advanced knowledge and skills. The idea of training with increasingly complex tasks, known as curriculum learning, has been applied in reinforcement learning when Selfridge et al. (1985) used a carefully curated sequence of tasks to train agents to solve a modified Cart Pole system. In recent years, there has been a growing interest in automatically generating curricula tailored to the agent's current capabilities, which opens up a new venue called "Automatic Curriculum Learning" (Portelas et al., 2020). An automatic curriculum generator requires two components: an encoded task space and a task characterization function (Narvekar et al., 2020; Wang et al., 2020). Task space encoding is often a bijective function that maps a task to a low dimensional vector (Parker-Holder et al., 2022; Klink et al., 2022; Florensa et al., 2018; Jiang et al., 2021; Portelas et al., 2019; Wang et al., 2019; 2020; Cho et al., 2023; Huang et al., 2022a). A proper task space encoding lays the foundation of a reasonable task characterization function measuring the fitness of tasks (Florensa et al., 2018; Dennis et al., 2020; Andreas et al., 2017; Sukhbaatar et al., 2018; Jiang et al., 2021). New training tasks, called source tasks, are generated by changing the target task's state space or parameters of transition functions in the encoded task space. A system designer then determines in which order the agent should be trained in these source tasks, following the task characterization function. The set of generated source tasks and the training order defined upon this set defines a *curriculum* for the learning agent. Please see App. G for more related work.

While impressive, most curriculum RL methods described so far rely on the assumption that generated source tasks are aligned with the target. Consequently, the agent could pick up some valuable skills by training in such source tasks, allowing it to behave optimally in certain situations in the target environment. However, this critical assumption does not necessarily hold in many real-world decision-making settings. For concreteness, consider a modified Sokoban game shown in Fig. 1 inspired by Schrader (2018) where an unobserved confounder $U_t$ randomly determines the box color $C_t$ (0 for yellow, 1 for blue) at every time step $t$. The agent receives a positive reward $Y_t$ only when it pushes the box to the goal state when the box color appears yellow ($U_t = 0$); otherwise,

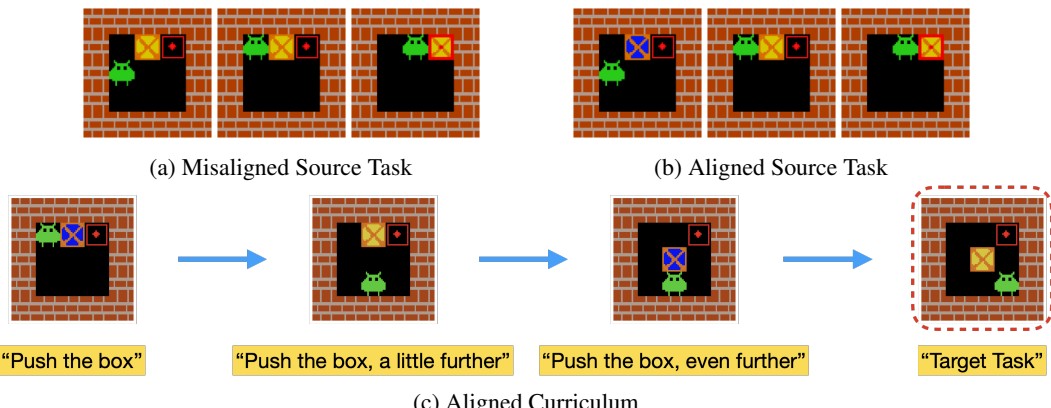

(a) Misaligned Source Task        (b) Aligned Source Task

"Push the box"    "Push the box, a little further"    "Push the box, even further"    "Target Task"

(c) Aligned Curriculum

Figure 1: Examples of (a) full episode of a misaligned source task that intervenes in the box color, (b) full episode of an aligned source task that only changes the initial box location, and (c) an aligned curriculum where none of the source tasks intervenes in the box's color.

it gets penalized ($U_t = 1$). We apply several state-of-the-art curriculum generators that construct source tasks by fixing the box color to yellow or blue, including ALP-GMM (Portelas et al., 2019), PLR (Jiang et al., 2021), Goal-GAN (Florensa et al., 2018), and Currot (Klink et al., 2022). Fig. 1a shows an example of the generated source tasks. We evaluate agents' performance trained by those generated curricula and compare it with the one directly trained in the target task. Surprisingly, simulation results shown in Fig. 2 reveal that agents trained by the curricula failed to learn to push the yellow box to the destination. This suggests source tasks generated by intervening in the box color are misaligned; that is, training in these source tasks harms the agents' target task performance.

Several observations follow from the Sokoban example. (1) A curriculum designer generates source tasks by modifying the data-generating mechanisms in the target tasks. (2) Such modifications could lead to a shift in system dynamics between the target task and source tasks. When this distribution shift is significant, training in source tasks may harm the agent's learning. (3) The agent must avoid misaligned source tasks to achieve optimal learning performance. There exist methods attempting to address the challenges of misaligned source tasks leveraging a heuristic similarity measure between the target and source

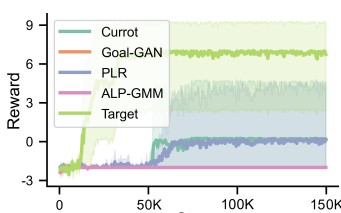

Figure 2: The average performance of curriculum generators.

tasks (Svetlik et al., 2017; Silva & Costa, 2018). Yet, a systematic and theoretically justified approach for exploiting other types of knowledge, e.g., qualitative, about the target task is missing.

This paper aims to address the challenges of misaligned source tasks in curriculum generation by exploring causal relationships among variables present in the underlying environment. To realize this, we formalize curriculum learning in the theoretical framework of structural causal models (SCMs) (Pearl, 2009). This formulation allows us to characterize misaligned source tasks by examining the structural invariance across the optimal policies obtained from the target and source tasks. More specifically, our contributions are summarized as follows. (1) We derive a sufficient graphical condition determining potentially misaligned source tasks. (2) We develop efficient algorithms for detecting misaligned source tasks and constructing source tasks that are guaranteed to align with the target task. (3) We introduce a novel augmentation procedure that enables state-of-the-art curriculum learning algorithms to generate aligned curricula to accelerate the agent's learning. Finally, we validate the proposed framework through extensive experiments in various decision-making tasks.

## 1.1 PRELIMINARIES

This section introduces necessary notations and definitions that will be used throughout the discussion. We use capital letters ($X$) to denote a random variable, lowercase letters ($x$) to represent a specific value of the random variable, and $\Omega(\cdot)$ to denote the domain of a random variable. We use bold capital letters ($\boldsymbol{V}$) to denote a set of random variables and use $|\boldsymbol{V}|$ to denote its cardinality.

The basic semantic framework of our analysis rests on *structural causal models* (SCMs) (Pearl, 2009; Bareinboim & Pearl, 2016). An SCM $\mathcal{M}$ is a tuple $\langle \boldsymbol{U}, \boldsymbol{V}, \mathscr{F}, P \rangle$, where $\boldsymbol{U}$ is a set of exogenous variables and $\boldsymbol{V}$ is a set of endogenous variables. $\mathscr{F}$ is a set of functions s.t. each $f_V \in \mathscr{F}$ decides values of an endogenous variable $V \in \boldsymbol{V}$ taking as argument a combination of other variables in the system. That is, $V \leftarrow f_V(\boldsymbol{PA}_V, \boldsymbol{U}_V), \boldsymbol{PA}_V \subseteq \boldsymbol{V}, \boldsymbol{U}_V \subseteq \boldsymbol{U}$. Values of exogenous variables $\boldsymbol{U}$ are drawn from the exogenous distribution $P(\boldsymbol{U})$. A policy $\pi$ over a subset of variables $\boldsymbol{X} \subseteq \boldsymbol{V}$ is a sequence of decision rules $\{\pi(X|\boldsymbol{S}_X)\}_{X \in \boldsymbol{X}}$, where every $\pi(X|\boldsymbol{S}_X)$ is a probability distribution mapping from domains of a set of covariates $\boldsymbol{S}_X \subseteq \boldsymbol{V}$ to the domain of action $X$. An intervention following a policy $\pi$ over variables $\boldsymbol{X}$, denoted by $\mathrm{do}(\pi)$, is an operation which sets values of every $X \in \boldsymbol{X}$ to be decided by policy $X \sim \pi(X|\boldsymbol{S}_X)$ (Correa & Bareinboim, 2020), replacing the functions $f_{\boldsymbol{X}} = \{f_X : \forall X \in \boldsymbol{X}\}$ that would normally determine their values. For an SCM $\mathcal{M}$, let $\mathcal{M}_\pi$ be a submodel of $M$ induced by intervention $\mathrm{do}(\pi)$. For a set $\boldsymbol{Y} \subseteq \boldsymbol{V}$, the interventional distribution $P(\boldsymbol{Y}; \pi)$ is defined as the distribution over $\boldsymbol{Y}$ in the submodel $\mathcal{M}_\pi$, i.e., $P_{\mathcal{M}}(\boldsymbol{Y}; \pi) \triangleq P_{\mathcal{M}_\pi}(\boldsymbol{Y})$; restriction $\mathcal{M}$ is left implicit when it is obvious.

Each SCM $\mathcal{M}$ is also associated with a causal diagram $\mathcal{G}$ (e.g., Fig. 3a), which is a directed acyclic graph (DAG) where nodes represent endogenous variables $\boldsymbol{V}$ and arrows represent the arguments $\boldsymbol{PA}_V, \boldsymbol{U}_V$ of each structural function $f_V \in \mathscr{F}$. Exogenous variables $\boldsymbol{U}$ are often not explicitly shown by convention. However, a bi-directed arrow $V_i \leftrightarrow V_j$ indicates the presence of an unobserved confounder (UC), $U_{i,j} \in \boldsymbol{U}$ affecting $V_i, V_j$, simultaneously (Bareinboim et al., 2022). We will use standard graph-theoretic family abbreviations to represent graphical relationships, such as parents ($pa$), children ($ch$), descendants ($de$), and ancestors ($an$). For example, the set of parent nodes of $\boldsymbol{X}$ in $\mathcal{G}$ is denoted by $pa(\boldsymbol{X})_{\mathcal{G}} = \cup_{X \in \boldsymbol{X}} pa(X)_{\mathcal{G}}$. Capitalized versions $Pa, Ch, De, An$ include the argument as well, e.g., $Pa(\boldsymbol{X})_{\mathcal{G}} = pa(\boldsymbol{X})_{\mathcal{G}} \cup \boldsymbol{X}$. A path from a node $X$ to a node $Y$ in $\mathcal{G}$ is a sequence of edges that does not include a particular node more than once. Two sets of nodes $\boldsymbol{X}, \boldsymbol{Y}$ are said to be d-separated by a third set $\boldsymbol{Z}$ in a DAG $\mathcal{G}$, denoted by $(\boldsymbol{X} \perp\!\!\!\perp \boldsymbol{Y} | \boldsymbol{Z})_{\mathcal{G}}$, if every edge path from nodes in $\boldsymbol{X}$ to nodes in $\boldsymbol{Y}$ is "blocked" by nodes in $\boldsymbol{Z}$. The criterion of blockage follows Pearl (2009, Def. 1.2.3). For more details on SCMs, we refer readers to Pearl (2009); Bareinboim et al. (2022). For the relationship between (PO)MDPs and SCMs, please see App. H.

## 2 Challenges of Misaligned Source Tasks

This section will formalize the concept of aligned source tasks and provide an efficient algorithmic procedure to find such tasks based on causal knowledge about the data-generating process. Formally, a planning/policy learning task (for short, a task) is a decision-making problem composed of an environment and an agent. We focus on the sequential setting where the agent determines values of a sequence of actions $\boldsymbol{X} = \{X_1, \ldots, X_H\}$ based on the input of observed states $\{\boldsymbol{S}_1, \ldots, \boldsymbol{S}_H\}$. The mapping between states and actions defines the space of candidate policies, namely,

**Definition 1** (Policy Space). For an SCM $\mathcal{M} = \langle \boldsymbol{U}, \boldsymbol{V}, \mathscr{F}, P \rangle$, a policy space $\Pi$ is a set of policies $\pi$ over actions $\boldsymbol{X} = \{X_1, \ldots, X_H\}$. Each policy $\pi$ is a sequence of decision rules $\{\pi_1(X_1|\boldsymbol{S}_1), \ldots, \pi_H(X_H|\boldsymbol{S}_H)\}$ where for every $i = 1, \ldots, H$,

(i) Action $X_i$ is a non-descendent of future actions $X_{i+1}, \ldots, X_H$, i.e., $X_i \in \boldsymbol{V} \setminus De(\bar{\boldsymbol{X}}_{i+1:H})$;

(ii) States $\boldsymbol{S}_i$ are non-descendants of future actions $X_i, \ldots, X_H$, i.e., $\boldsymbol{S}_i \subseteq \boldsymbol{V} \setminus De(\bar{\boldsymbol{X}}_{i:H})$.

Henceforth, we will consistently denote such a policy space by $\Pi = \{\langle X_1, \boldsymbol{S}_1 \rangle, \ldots, \langle X_H, \boldsymbol{S}_H \rangle\}$.

The agent interacts with the environment by performing intervention $\mathrm{do}(\pi), \forall \pi \in \Pi$ to optimize a reward function $\mathcal{R}(\boldsymbol{Y})$ taking a set of reward signals $\boldsymbol{Y} \subseteq \boldsymbol{V}$ as input.[1] A policy space, a reward function, and an SCM environment formalize a target decision-making task. We will graphically describe a target task using an augmented causal diagram $\mathcal{G}$ constructed from the SCM $\mathcal{M}$; actions $\boldsymbol{X}$ are highlighted in blue; reward signals $\boldsymbol{Y}$ are highlighted in red; input states $\boldsymbol{S}_i$ for every action $X_i \in \boldsymbol{X}$ are shaded in light blue. For instance, Fig. 3a shows a causal diagram representing the decision-making task in the Sokoban game (Fig. 1). For every time step $i = 1, \ldots, H$, $L_i$ stands for the agent's location, $B_i$ for the box location, and $C_i$ for the box color.

---

[1] For instance, a cumulative discounted reward is defined as $\mathcal{R}(\boldsymbol{Y}) = \sum_{i=1}^{H} \gamma^{i-1} Y_i$ where $Y_i \in \boldsymbol{V}$, $i = 1, \ldots, H$, are endogenous variables, and $\gamma \in (0, 1]$ is a discount factor.

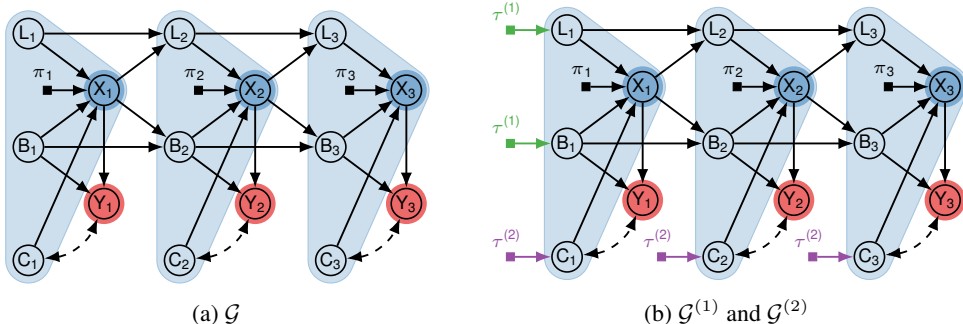

(a) $\mathcal{G}$            (b) $\mathcal{G}^{(1)}$ and $\mathcal{G}^{(2)}$

Figure 3: Causal diagram for (a) the target task $\mathcal{T}$; and (b) comparing domain discrepancies between the target task $\mathcal{T}$ and source tasks $\mathcal{T}^{(1)}$ and $\mathcal{T}^{(2)}$. (b) is (a) augmented by edit indicators.

**Definition 2** (Target Task). A target task is a tuple $\mathcal{T} = \langle \mathcal{M}, \Pi, \mathcal{R} \rangle$, where $\mathcal{M} = \langle \boldsymbol{U}, \boldsymbol{V}, \mathscr{F}, P \rangle$ is an SCM, $\Pi$ is a policy space over actions $\boldsymbol{X} \subseteq \boldsymbol{V}$, and $\mathcal{R}$ is a reward function over signals $\boldsymbol{Y} \subseteq \boldsymbol{V}$.

The goal is to find an optimal policy $\pi^* \in \Pi$ that maximizes the expected reward function $\mathbb{E}[\mathcal{R}(\boldsymbol{Y}); \pi]$ evaluated in the underlying environment $\mathcal{M}$, i.e.,

$$\pi^* = \arg\max_{\pi \in \Pi} \mathbb{E}_{\mathcal{M}}[\mathcal{R}(\boldsymbol{Y}); \pi]. \tag{1}$$

When the detailed parametrization of the SCM $\mathcal{M}$ is provided, the optimal policy $\pi^*$ is obtainable by applying planning algorithms, e.g., dynamic programming (Bellman, 1966) or influence diagrams (Koller & Milch, 2003). However, when underlying system dynamics are complex or the state-action domains are high-dimensional, it might be challenging to solve an optimal policy even with state-of-the-art planning algorithms. We will then consider the curriculum learning approach (Selfridge et al., 1985), where the agent is not immediately trained in the target task but provided with a sequence of related yet simplified source tasks.

**Definition 3** (Source Task). For a target task $\mathcal{T} = \langle \mathcal{M}, \Pi, \mathcal{R} \rangle$, a source task $\mathcal{T}^{(j)}$ is a tuple $\langle \mathcal{M}^{(j)}, \Pi, \mathcal{R}, \Delta^{(j)} \rangle$ where $\mathcal{M}^{(j)}$ is an SCM compatible with the same causal diagram as $\mathcal{M}$, i.e., $\mathcal{G}_{\mathcal{M}} = \mathcal{G}_{\mathcal{M}^{(j)}}$; a set of variables $\Delta^{(j)} \subseteq \boldsymbol{V}$ is called *edits* where there might exist a discrepancy that $f_V \neq f_V^{(j)}$ or $P(\boldsymbol{U}_V) \neq P^{(j)}(\boldsymbol{U}_V)$ for every $V \in \Delta^{(j)}$.

In practice, source tasks are constructed from the target task by modifying parameters of the underlying structural functions $\mathscr{F}$ or exogenous distributions $P(\boldsymbol{U})$. Consider again the Sokoban game described in Fig. 1. The system designer could generate a source task $\mathcal{T}^{(1)}$ by changing the agent and box's initial location $L_1, B_1$. Fig. 3b shows a causal diagram $\mathcal{G}^{(1)}$ [2] representing the source task $\mathcal{T}^{(1)}$; $\tau^{(1)}$ is an *edit indicator* representing the domain discrepancies $\Delta^{(1)}$ between the target $\mathcal{T}$ and source tasks $\mathcal{T}^{(1)}$. Here, arrows $\tau^{(1)} \rightarrow L_1, \tau^{(1)} \rightarrow B_1$ suggest that structural functions $f_{L_1}, f_{B_1}$ or exogenous distributions $P(\boldsymbol{U}_{L_1}, \boldsymbol{U}_{B_1})$ have been changed in the source task $\mathcal{T}^{(1)}$ while other parts of the system remain the same as the target task $\mathcal{T}$.

By simplifying the system dynamics, learning an optimal policy in the source task $\mathcal{T}^{(j)}$ could be easier than in the target task $\mathcal{T}$. The expectation here is that the optimal decision rules $\pi^{(j)}$ over some actions $\boldsymbol{X}^{(j)} \subseteq \boldsymbol{X}$ remain invariant across the source and target tasks. If so, we will call such source tasks as *aligned*. Training in an aligned source task thus guides the agent to move toward an optimal policy $\pi^*$. For example, Fig. 1b shows an aligned source task for the Sokoban game where the agent and box's locations are set close to the goal state. By training in the simplified task, the agent learns the optimal decision rule to push the yellow box to the goal state in this game.

However, modifying the target task could lead to a misaligned source task whose system dynamics differ significantly from the target. Interestingly and more seriously, training in these source tasks may "harm" the agent's performance, resulting in suboptimal decision rules, as illustrated next.

---

[2]We will consistently use the superscript $(j)$ to indicate a diagram $\mathcal{G}^{(j)} \triangleq \mathcal{G}_{\mathcal{M}^{(j)}}$ associated with a source task $\mathcal{T}^{(j)}$. Similarly, we write $P^{(j)}(\boldsymbol{Y}; \pi) = P_{\mathcal{M}^{(j)}}(\boldsymbol{Y}; \pi)$ and $\pi^{(j)} = \arg\max_{\pi \in \Pi} \mathbb{E}_{\mathcal{M}^{(j)}}[\mathcal{R}(\boldsymbol{Y}); \pi]$.

**Example 1** (Misaligned Source Task). Consider the Sokoban game $\mathcal{T} = \langle \mathcal{M}, \Pi, \mathcal{R} \rangle$ described in Fig. 1; Fig. 3a shows its causal diagram $\mathcal{G}$. Specifically, the box color $C_i$ (0 for yellow, 1 for blue) is determined by an unobserved confounder $U_i \in \{0, 1\}$ randomly drawn from a distribution $P(U_i = 1) = 3/4$. Box location $B_i$ and agent location $L_i$ are determined following system dynamics in deterministic grid worlds (Chevalier-Boisvert et al., 2018). The reward signal $Y_i$ is given by,

$$Y_i = \begin{cases} 10 & \text{if } B_i = \text{``next to goal''} \wedge X_i = \text{``push''} \wedge (U_i = 0) \\ -10 & \text{if } B_i = \text{``next to goal''} \wedge X_i = \text{``push''} \wedge (U_i = 1) \\ -0.1 & \text{otherwise} \end{cases} \tag{2}$$

If the agent pushes the box into the goal location (top right corner in Fig. 1), it receives a positive reward when the box appears yellow; it gets penalized when the box appears blue. Since $C_i \leftarrow U_i$, evaluating the conditional reward $\mathbb{E}\left[Y_i \middle| b_i, c_i; \mathrm{do}(x_i)\right]$ in the Sokoban environment $\mathcal{M}$ gives,

$$\mathbb{E}\left[Y_i \middle| B_i = \text{``next to goal''}, C_i; \mathrm{do}\left(X_i = \text{``push''}\right)\right] = \begin{cases} 10 & \text{if } C_i = 0 \\ -10 & \text{if } C_i = 1 \end{cases} \tag{3}$$

Thus, the agent should aim to push yellow boxes to the goal location in the target. The curriculum designer now attempts to generate a source task $\mathcal{T}^{(2)}$ by fixing the box color to yellow, i.e., $C_i \leftarrow 0$. Fig. 3b shows the causal diagram $\mathcal{G}^{(2)}$ associated with the source environment $\mathcal{M}^{(2)}$ where edit indicators $\tau^{(2)}$ denote the change in the structural function $f_{C_i}$ determining the box color $C_i$. Evaluating the conditional reward $\mathbb{E}\left[Y_i \middle| b_i; \mathrm{do}(c_i, x_i)\right]$ in this manipulated environment $\mathcal{M}^{(2)}$ gives

$$\mathbb{E}^{(2)}\left[Y_i \middle| B_i = \text{``next to goal''}; \mathrm{do}\left(C_i = 0, X_i = \text{``push''}\right)\right] = -5. \tag{4}$$

Detailed computations are provided in App. B. Perhaps counter-intuitively, pushing the yellow box to the goal location in the source task $\mathcal{T}^{(2)}$ results in a negative expected reward. This is because box color $C_i$ is only a proxy to the unobserved $U_i$ that controls the reward. Fixing $C_i$ won't affect $Y$ but only breaks this synergy, hiding the critical information of $U_i$ from the agent. Consequently, when training in the source task $\mathcal{T}^{(2)}$, the agent will learn to *never push* the box even when it is next to the goal location, which is suboptimal in the target Sokoban game $\mathcal{T}$. ∎

## 2.1 CAUSALLY ALIGNED SOURCE TASK

Example 1 suggests that naively training in a misaligned source task may lead to suboptimal performance in the target task. The remainder of this section will introduce an efficient strategy to avoid misaligned source tasks, provided with the causal knowledge of the underlying data-generating mechanisms in the environment. For a target task $\mathcal{T} = \langle \mathcal{M}, \Pi, \mathcal{R} \rangle$, let $\mathcal{G}$ be the causal diagram associated with $\mathcal{M}$. Let $\mathcal{G}_\pi$ be an *intervened* diagram obtained from $\mathcal{G}$ by replacing incoming arrows if action $X_i \in \boldsymbol{X}$ with arrows from input states $\boldsymbol{S}_i$ for every action $X_i \in \boldsymbol{X}$. We first characterize a set of variables $\Delta^{(j)} \subseteq \boldsymbol{V}$ amenable to editing (for short, editable states) using independence relationships between edit indicators $\tau^{(j)}$ and reward signals $\boldsymbol{Y}$. Formally,

**Definition 4** (Editable States). For a target task $\mathcal{T} = \langle \mathcal{M}, \Pi, \mathcal{R} \rangle$, let $\mathcal{G}$ be a causal diagram of $\mathcal{M}$ and $\boldsymbol{X}^{(j)} \subseteq \boldsymbol{X}$ be a subset of actions. A set of variables $\Delta^{(j)} \subseteq \boldsymbol{V} \setminus \boldsymbol{X}^{(j)}$ is *editable* w.r.t $\boldsymbol{X}^{(j)}$ if and only if $\forall X_i \in \boldsymbol{X}^{(j)}$, the following independence holds in the intervened diagram $\mathcal{G}_\pi$,

$$\left(\tau^{(j)} \perp\!\!\!\perp \boldsymbol{Y} \cap De(X_i) \mid X_i, \boldsymbol{S}_i\right), \tag{5}$$

where $\tau^{(j)}$ is the set of added edit indicators pointing into nodes in $\Delta^{(j)}$.

For example, consider again the Sokoban game described in Example 1. The initial agent and box's position $\Delta^{(1)} = \{B_1, L_1\}$ is editable with regard to all actions $\boldsymbol{X}$ following Def. 4. Precisely, in the augmented diagram $\mathcal{G}^{(1)}$ of Fig. 3b, for every action $X_i \in \boldsymbol{X}$, the edit indicators $\tau^{(1)}$ are d-separated the reward signals $\boldsymbol{Y} \cap De(X_i) = \{Y_i, \ldots, Y_H\}$ given input states $\{L_i, B_i, C_i\}$. On the other hand, the set of box color variables $\Delta^{(2)} = \{C_1, \ldots, C_H\}$ are not editable w.r.t. actions $\boldsymbol{X}$ since in the augmented diagram $\mathcal{G}^{(2)}$ of Fig. 3b, for every action $X_i \in \boldsymbol{X}$, there exists an active path between edit indicators $\tau^{(2)}$ and reward signals $\{Y_i, \ldots, Y_H\}$ given action $X_i$ and input states $\{L_i, B_i, C_i\}$, violating the criterion given by Def. 4.

For a fixed policy $\pi \in \Pi$, for any subset $\boldsymbol{S} \subseteq \boldsymbol{V}$, we denote by $\Omega^{(j)}(\boldsymbol{S}; \pi) = \{\forall \boldsymbol{s} \in \Omega(\boldsymbol{S}) \mid P_{\mathcal{M}}(\boldsymbol{s}; \pi) > 0\}$ the set of *reachable* values of $\boldsymbol{S}$, which is the set of states that are possible to reach in a source task $\mathcal{T}^{(j)}$ under intervention $do(\pi)$. The following proposition establishes that modifying functions and distributions over a set of editable states $\Delta^{(j)}$ leads to an aligned source task.

**Theorem 1** (Causally Aligned Source Task). *For a target task $\mathcal{T} = \langle \mathcal{M}, \Pi, \mathcal{R} \rangle$, let $\mathcal{T}^{(j)} = \langle \mathcal{M}^{(j)}, \Pi, \mathcal{R}, \Delta^{(j)} \rangle$ be a source task of $\mathcal{T}$ by modifying states $\Delta^{(j)} \subseteq \boldsymbol{V}$. If $\Delta^{(j)}$ is editable w.r.t some actions $\boldsymbol{X}^{(j)} \subseteq \boldsymbol{X}$, then for every action $X_i \in \boldsymbol{X}^{(j)}$,*

$$\pi_i^*(X_i \mid \boldsymbol{s}_i) = \pi_i^{(j)}(X_i \mid \boldsymbol{s}_i), \ \ \forall \boldsymbol{s}_i \in \Omega^{(j)}(\boldsymbol{S}_i; \pi^{(j)}) \cap \Omega(\boldsymbol{S}_i; \pi^*) \tag{6}$$

*where $\pi^*, \pi^{(j)} \in \Pi$ are optimal policies in the target $\mathcal{T}$ and source $\mathcal{T}^{(j)}$ tasks, respectively.*

Thm. 1 implies that whenever states $\Delta^{(j)}$ is editable w.r.t. some actions $\boldsymbol{X}^{(j)}$, one could always construct an aligned source task $\mathcal{T}^{(j)}$ such that the optimal decision rules $\pi^*$ over $\boldsymbol{X}^{(j)}$ is invariant across the target $\mathcal{T}$ and source $\mathcal{T}^{(j)}$ tasks. Consequently, one could *transport* these optimal decision rules trained in the source task $\mathcal{T}^{(j)}$ without harming the agent's performance in the target domain $\mathcal{T}$. [3] For example, in the Sokoban game of Example 1, since initial states $\Delta^{(1)} = \{B_1, L_1\}$ is editable w.r.t. actions $\boldsymbol{X}$, moving the agent and box's location leads to an aligned source task, which allows the agent to learn how to behave optimally when getting closer to the goal state. However, the performance guarantee in Thm. 1 does not necessarily hold when states $\Delta^{(j)}$ are not editable. For instance, recall that $\Delta^{(2)} = \{C_1, \ldots, C_H\}$ are not editable in the Sokoban game. Modifying the box's color could lead to a misaligned source task $\mathcal{T}^{(2)}$. An agent trained in this source task could pick up undesirable behaviors, as demonstrated in Example 1.

Algo. 1 describes an algorithmic procedure, FINDMAXEDIT, to find a maximal editable set $\Delta^{(j)}$ in a causal diagram $\mathcal{G}$ w.r.t. a set of actions $\boldsymbol{X}^{(j)} \subseteq \boldsymbol{X}$. A set of editable states $\Delta^{(j)}$ is maximal w.r.t. $\boldsymbol{X}^{(j)}$ if there is no other editable states $\Delta_*^{(j)}$ strictly containing $\Delta^{(j)}$. We always prefer a maximal editable set since it offers the maximum freedom to simplify the system dynamics in the target task. Particularly, FINDMAXEDIT it-

---

**Algorithm 1:** FINDMAXEDIT

**Input:** A causal diagram $\mathcal{G}_\pi$ and a set of actions, $\boldsymbol{X}^{(j)}$.
**Output:** The maximal editable states $\Delta^{(j)}$ w.r.t $\boldsymbol{X}^{(j)}$.
Let $\Delta^{(j)} \leftarrow \emptyset$;
**for** $V \in \boldsymbol{V} \setminus \boldsymbol{X} \cup \boldsymbol{Y}$ **do**
    $\Delta^{(j)} \leftarrow \Delta^{(j)} \cup \{V\}$;
    **for** *every* $X_i \in \boldsymbol{X}^{(j)}$ **do**
        **if** $\left(\tau^{(j)} \not\perp \boldsymbol{Y} \cap De(X_i) \mid X_i, \boldsymbol{S}_i\right)$ *in* $\mathcal{G}_\pi$ **then**
            Remove $V$ from $\Delta^{(j)}$ and break;
**return** $\Delta^{(j)}$;

---

eratively adds endogenous variables $\boldsymbol{V} \setminus (\boldsymbol{X} \cup \boldsymbol{Y})$ to the editable states $\Delta^{(j)}$ and test the independence criterion in Def. 4. This procedure continues until it cannot add any more endogenous variables. Evidently, FINDMAXEDIT returns a maximal editable set $\Delta^{(j)}$ w.r.t. $\boldsymbol{X}^{(j)}$. A natural question arising at this point is whether the ordering of endogenous variables $V$ changes the output. Fortunately, the next proposition shows that this is not the case.

**Theorem 2.** *For a target task $\mathcal{T} = \langle \mathcal{M}, \Pi, \mathcal{R} \rangle$, let $\mathcal{G}_\pi$ be an intervened causal diagram of $\mathcal{M}$ and let $\boldsymbol{X}^{(j)} \subseteq \boldsymbol{X}$ be a subset of actions. FINDMAXEDIT $\left(\mathcal{G}_\pi, \boldsymbol{X}^{(j)}\right)$ returns a maximal editable set $\Delta^{(j)}$ w.r.t actions $\boldsymbol{X}^{(j)}$; moreover, such a maximal set $\Delta^{(j)}$ is unique.*

Let $n$ and $m$ denote the number of nodes and edges in the intervened diagram $\mathcal{G}_\pi$ and let $d$ be the number of actions $\boldsymbol{X}$. Since testing d-separation has a time complexity of $\mathcal{O}(n + m)$, FIND-MAXEDIT has a time complexity of $\mathcal{O}(d(n + m))$. We also provide other algorithmic procedures for directly deciding a set's editability and constructing editable sets for a target task $\mathcal{T}$ in App. C.

## 3 CURRICULUM LEARNING VIA CAUSAL LENS

Once a collection of source tasks is constructed, the system designer could organize them into an ordered list, called a *curriculum*, as defined next:

---

[3]Causal aligned source tasks (Thm. 1) and editable states (Def. 4) are related to the concept of *transportability* in causal inference literature (Bareinboim & Pearl, 2016), which generalizes estimation of unknown causal effects from different domains. Here we study the generalizability of an optimal decision policy.

**Definition 5** (Curriculum). For a target task $\mathcal{T} = \langle \mathcal{M}, \Pi, \mathcal{R} \rangle$, a curriculum $\mathcal{C}$ for $\mathcal{T}$ is a sequence of source tasks $\{\mathcal{T}^{(j)}\}_{j=1}^N$, where $\mathcal{T}^{(j)} = \langle \mathcal{M}^{(j)}, \Pi, \mathcal{R}, \Delta^{(j)} \rangle$.

For instance, Fig. 1c describes a curriculum in the Sokoban game where the agent and the box are placed increasingly further away from the goal location. Given a curriculum $\mathcal{C}$, a typical curriculum learning algorithm trains the agent sequentially in each source task, following the curriculum's ordering. Algo. 2 shows the pseudo-code describing this training process. It first initializes an arbitrary baseline policy $\pi^{(0)}$. For every source task $\mathcal{T}^{(j)} \in \mathcal{C}$, the algorithm updates the current

---

**Algorithm 2:** CURRICULUM LEARNING

**Input:** A curriculum $\mathcal{C}$.
**Output:** A policy $\pi^{(N)} \in \Pi$.
Initialize a baseline policy $\pi^{(0)}$;
**for** $j = 1, ..., N$ **do**
> Update a policy $\pi^{(j)}$ from $\pi^{(j-1)}$ such that
> $$\pi^{(j)} = \arg\max_{\pi \in \Pi} \mathbb{E}_{\mathcal{M}^{(j)}} \left[ \mathcal{R}(\boldsymbol{Y}); \pi \right] \quad (7)$$

**return** $\pi^{(N)}$;

---

policy $\pi^{(j-1)}$ such that the new policy $\pi^{(j)}$ is optimal in the source task $\mathcal{T}^{(j)}$. This step could be performed using a standard gradient-based algorithm, e.g., the policy gradient (Sutton & Barto, 2018). The expectation is that, as the agent picks up more skills in the source tasks, it could consistently improve its performance in the target task or at least not regress.

**Definition 6** (Causally Aligned Curriculum). For a target task $\mathcal{T} = \langle \mathcal{M}, \Pi, \mathcal{R} \rangle$, let $\mathcal{C} = \{\mathcal{T}^{(j)}\}_{j=1}^N$ be a curriculum for $\mathcal{T}$. Curriculum $\mathcal{C}$ is said to be causally aligned with $\mathcal{T}$ if for every $j = 1, \ldots, N-1$, the set of invariant optimal decision rules across the source task and the target task expands, i.e.,

$$\left( \pi^{(j)} \cap \pi^* \right) \subseteq \left( \pi^{(j+1)} \cap \pi^* \right), \tag{8}$$

where $\pi^* \in \Pi$ is an optimal policy in the target task $\mathcal{T}$.

A naive approach to construct a causally aligned curriculum is to (1) construct a set of aligned source tasks by modifying editable states (Thm. 1) and (2) organize these tasks in an arbitrary ordering. However, the following example shows this is not a viable option.

**Example 2** (Overwriting in Curriculum Learning). Consider a two-stage target task where action $X_1$ takes input $H$ and $X_2$ takes input $Z$. The task SCM is, $H = U_H, Z = \neg X_1 \oplus U_Z, Y_1 = 0.5 * (H \oplus X_1), Y_2 = \neg H \oplus X_2 \wedge Z$ where $P(U_Z = 1) = 1/2, P(U_H = 1) = 1/10$. Other than the reward $Y_1, Y_2$, all other variables are binary. The optimal policy for the target task is $\pi^*(X_1 = \neg H | H) = 1$, $\pi^*(X_2 = 0 | Z) = 1$. We create two source tasks. For $\mathcal{T}^{(1)}$, let $P(U_H = 1) = 9/10$ while other parts stay the same as target task $\mathcal{T}$. For $\mathcal{T}^{(2)}$, let $Z = \neg X_1$ while other parts stay the same as target task $\mathcal{T}$. From the causal diagram, we see that $\Delta^{(1)} = \{H\}$ is editable w.r.t $\boldsymbol{X}^{(1)} = \{X_1\}$ and $\Delta^{(2)} = \{Z\}$ is editable w.r.t $\boldsymbol{X}^{(2)} = \{X_2\}$.

Now if the agent is trained in a curriculum $\mathcal{C} = \{\mathcal{T}^{(1)}, \mathcal{T}^{(2)}\}$, its target task performance will deteriorate instead of improving. To witness, the optimal policy for $X_2$ in $\mathcal{T}^{(1)}$ is $\pi^{(1)}(X_2 = 1 | Z) = 1$ and the optimal policy for $X_1$ in $\mathcal{T}^{(2)}$ is $\pi^{(2)}(X_1 = 0 | H) = 1$. After training in $\mathcal{T}^{(1)}$, $\pi^{(1)}$ has an expected target task reward of 0.55 since $\pi^{(1)}(X_2 = 1 | Z)$ is not optimal in the target yet. So, the agent proceeds to train in $\mathcal{T}^{(2)}$. It will learn the optimal target policy for $X_2$, $\pi^*(X_2 = 0 | Z) = 1$. But in the mean time, optimal policy of $X_1$ learned from $\mathcal{T}^{(1)}$, $\pi^*(X_1 = \neg H | H) = 1$, is also overwritten by $\pi^{(2)}$. The agent will only receive 0.5 in the target task, which is even worse than before training in $\mathcal{T}^{(2)}$. This suggests that curriculum $\mathcal{C}$ is not causally aligned. ∎

In the above example, the agent fails to learn an optimal policy due to "policy overwriting". Fig. 4 provides a graphical representation of this phenomenon. Particularly, each source task $\mathcal{T}^{(1)}, \mathcal{T}^{(2)}$ covers one of the optimal decision rules over action $X_1, X_2$, respectively. An agent trained in one of the source tasks, say $\mathcal{T}^{(1)}$, learns the optimal decision rule $\pi_1^*$ for action $X_1$, but forgets the decision rule $\pi_2^*$ for the other action $X_2$ learned previously in $\mathcal{T}^{(2)}$. The same overwriting also occurs when the agent moves from task $\mathcal{T}^{(2)}$ to $\mathcal{T}^{(1)}$. This means that regardless of how the system designer orders the curriculum, e.g., $\mathcal{C} = \{\mathcal{T}^{(1)}, \mathcal{T}^{(2)}, \mathcal{T}^{(1)}, \mathcal{T}^{(2)}, \ldots\}$, the agent will always forget useful skills it picked up from

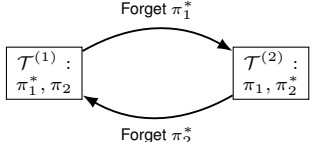

Figure 4: Policy overwriting described in Example 2.

previous source tasks, thus making it unable to achieve satisfactory performance in the target task. This example implies that there are more conditions for a curriculum to be "causally aligned".

## 3.1 Designing Causally Aligned Curriculum

We will next introduce a novel algorithmic procedure to construct a causally aligned curriculum while avoiding the issue of overwriting. We will focus on a general class of *soluble* target tasks, which generalizes the *perfect recall* criterion (Koller & Friedman, 2009) in the planning/decision-making literature (Lauritzen & Nilsson, 2001).

**Definition 7** (Soluble Target Task). A target task $\mathcal{T} = \langle \mathcal{M}, \Pi, \mathcal{R} \rangle$ is soluble if whenever $j < i$, $(\boldsymbol{Y} \cap De(X_i) \perp\!\!\!\perp \pi_j | \boldsymbol{S}_i, X_i)$ in $\mathcal{G}_\pi$, where $\pi_j$ is a newly added regime node pointing to $X_j$.

In words, Def. 7 says that for a soluble target task $\mathcal{T}$, for every action $X_i \in \boldsymbol{X}$, the input states $\boldsymbol{S}_i$ summarizes all the states and actions' history $\boldsymbol{S}_1, \ldots, \boldsymbol{S}_{i-1}, X_1, \ldots, X_{i-1}$. If this is the case, an optimal policy $\pi^*$ for task $\mathcal{T}$ is obtainable by solving a series of dynamic programs (Lauritzen & Nilsson, 2001; Koller & Milch, 2003). For instance, the Sokoban game $\mathcal{T}$ graphically described in Fig. 3a is soluble. For every time step $i = 1, \ldots, H$, given input states $\boldsymbol{S}_i = \{L_i, B_i, C_i\}$ and action $X_i$, regime variables $\pi_1, \ldots, \pi_{i-1}$ are d-separated from subsequent reward signals $Y_i, \ldots, Y_H$.

**Theorem 3** (Causally Aligned Curriculum). *For a soluble target task* $\mathcal{T} = \langle \mathcal{M}, \Pi, \mathcal{R} \rangle$*, a curriculum* $\mathcal{C} = \{\mathcal{T}^{(j)}\}_{j=1}^N$ *is causally aligned if the following conditions hold,*

*(i) Every source task* $\mathcal{T}^{(j)} \in \mathcal{C}$ *is causally aligned w.r.t. actions* $\boldsymbol{X}^{(j)}$ *(Def. 4);*
*(ii) For every* $j = 1, \ldots, N - 1$*, actions* $\boldsymbol{X}^{(j)} \subseteq \boldsymbol{X}^{(j+1)}$*.*

Consider again the Sokoban game described in Fig. 3a. Let $\mathcal{C} = \{\mathcal{T}^{(j)}\}_{j=1}^H$ be a curriculum such that for every source task $\mathcal{T}^{(j)}$ is obtained by editing the agent and box's location $\Delta^{(j)} = \{L_i, B_i\}$ at time step $i = H - j + 1$. We now examine conditions in Thm. 3 and see if $\mathcal{C}$ is causally aligned. First, Condition (i) holds since every source task $\mathcal{T}^{(j)}$ is causally aligned w.r.t. actions $\boldsymbol{X}^{(j)} = \{X_{H-j+1}, \ldots, X_H\}$ following discussion in the previous section. Also, Condition (ii) holds since for every $j = 1, \ldots, H - 1$, actions $\boldsymbol{X}^{(j)} \subseteq \boldsymbol{X}^{(j+1)}$. This implies that one could construct a causally aligned curriculum in the Sokoban game by repeatedly editing the agent and box' location following a reversed topological ordering; Fig. 1c describes such an example.

The idea in Thm. 3 suggests a natural procedure for constructing a causally aligned curriculum, which is implemented in FINDCAUSALCUR-RICULUM (Algo. 3). Particularly, it assumes access to a curriculum generator $\text{GEN}(\mathcal{T}, \Delta^{(j)})$ which generates a source task $\mathcal{T}^{(j)}$ by editing a set of states $\Delta^{(j)} \subseteq \boldsymbol{V}$ in the target task $\mathcal{T}$. It follows a reverse topological ordering over actions $\boldsymbol{X} = \{X_1, \ldots, X_H\}$. For every step $j = H, \ldots, 1$, the algorithm call the sub-routine FINDMAXEDIT (Algo. 1) to find a set of editable states $\Delta^{(j)}$ w.r.t. actions $\boldsymbol{X}^{(j)} =$

---

**Algorithm 3:** FINDCAUSALCURRICULUM

**Input:** A target task $\mathcal{T}$, a causal diagram $\mathcal{G}_\pi$
**Output:** A causally aligned curriculum $\mathcal{C}$
Let $\mathcal{C} \leftarrow \emptyset$;
**for** $j = H, \ldots, 1$ **do**
    Let $\boldsymbol{X}^{(j)} \leftarrow \{X_j, \ldots, X_H\}$;
    Let $\Delta^{(j)} \leftarrow \text{FINDMAXEDIT}(\mathcal{G}_\pi, \boldsymbol{X}^{(j)})$;
    Let $\mathcal{T}^{(j)} \leftarrow \text{GEN}(\mathcal{T}, \Delta^{(j)})$;
    Let $\mathcal{C} = \mathcal{C} \cup \{\mathcal{T}^{(j)}\}$;
**return** $\mathcal{C}$;

---

$\{X_j, \ldots, X_H\}$. It then calls the generator GEN to generate a source task $\mathcal{T}^{(j)}$ by editing states $\Delta^{(j)}$. The conditions in Thm. 3 ensure that Algo. 3 returns a causally aligned curriculum.

**Corollary 1.** *For a soluble target task* $\mathcal{T} = \langle \mathcal{M}, \Pi, \mathcal{R} \rangle$*, let* $\mathcal{G}_\pi$ *be an intervened causal diagram of* $\mathcal{M}$*.* FINDCAUSALCURRICULUM $(\mathcal{T}, \mathcal{G}_\pi)$ *returns a causally aligned curriculum.*

A more detailed discussion on the additional conditions under which a combination of Algs. 2 and 3 is guaranteed to find an optimal target task policy is provided in App. D.

## 4 Experiments

In this section, we build on Algo. 3 and different curriculum generators to evaluate causally aligned curricula for solving challenging tasks in which confounding bias is present and previous, non-causal generators cannot solve. In particular, we test four best-performing curriculum generators:

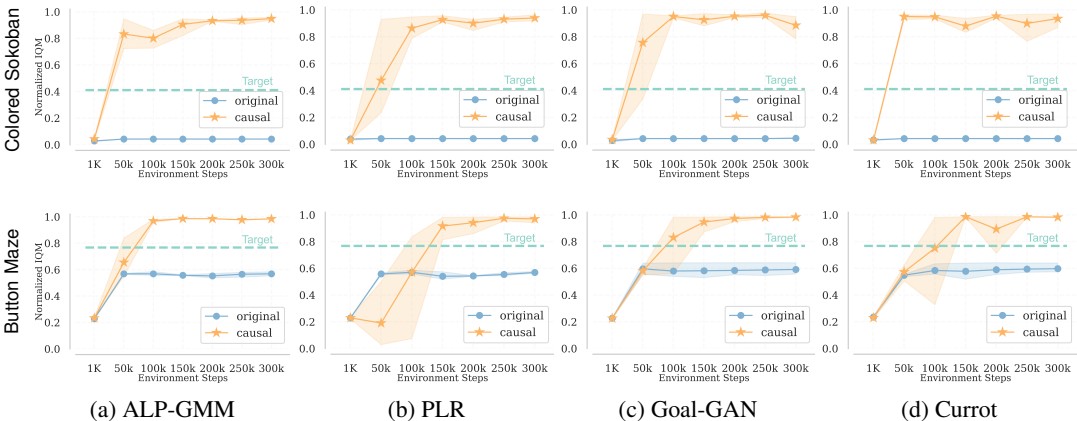

(a) ALP-GMM      (b) PLR      (c) Goal-GAN      (d) Currot

Figure 5: Target task performance of the agents at different training stages in Colored Sokoban (Row 1) and Button Maze (Row 2) using different curriculum generators (Columns). The horizontal green line shows the performance of the agent trained directly in the target. "original" refers to the unaugmented curriculum generator and "causal" refers to its causally augmented version.

ALP-GMM (Portelas et al., 2019), PLR (Jiang et al., 2021), Goal-GAN (Florensa et al., 2018), and Currot (Klink et al., 2022) in two confounded environments with pixel observations: (a) Colored Sokoban, (b) Button Maze, (c) Continuous Button Maze (App. F). All experiments are conducted with five random seeds and reported in Interquartile Mean (IQM) normalized w.r.t the minimum and maximum rewards with 95% confidence intervals shown in shades. See App. F for more details.

**Colored Sokoban.** Consider the same Sokoban game as shown in Example 1. The curriculum generators are allowed to vary the initial location of the agent, to vary the initial box location, and to intervene the box's color. Without editing, the box color syncs with the true underlying rewards, i.e., pushing a yellow box always yields a positive reward. However, after intervening the box color, this sync is broken and the agent has no information on the right time to push the box. As shown in Fig. 5, agents trained by original curriculum generators failed to converge due to this. After causal augmentation, those misaligned source tasks with intervened box color are all eliminated from the search space during curriculum generation. The causal versions of those generators successfully train the agent to converge efficiently and surpass those trained directly in the target task.

**Button Maze.** In this grid world environment (Chevalier-Boisvert et al., 2018), the agent must navigate to the goal location and step onto it at the right time. Specifically, after pushing the button, the goal region will turn green and yield a positive reward if the agent steps onto it. However, before pushing the button, there is only a 20% chance the agent gets a positive reward for reaching the goal, and the goal randomly blinks between red and green, independent of the underlying rewards. Curriculum generators can intervene the goal color and vary the agent's initial location but intervening goal colors creates misaligned curricula (Thm. 3). As shown in Fig. 5, agents trained by vanilla curriculum generators failed to learn at all, while the agents trained by their causally-augmented versions all converged to the optimal, even surpassing the one trained directly in the target task.

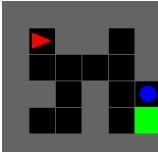

Figure 6: Button Maze.

## 5 CONCLUSION

We develop a formal treatment for automatic curriculum design in confounded environments through causal lenses. We propose a sufficient graphical criterion that edits must conform with to generate causally aligned source tasks in which the agent is guaranteed to learn optimal decision rules for the target task. We also develop a practical implementation of our graphical criteria, i.e., FIND-MAXEDIT, that augments the existing curriculum generators into ones that generate aligned source tasks regardless of the existence of unobserved confounders. Finally, we analyze causally aligned curricula' design principles with theoretical performance guarantees. The effectiveness of our approach is empirically verified in two high-dimensional pixel-based tasks.

## 6 ACKNOWLEDGEMENTS

This research was supported in part by the NSF, ONR, AFOSR, DoE, Amazon, JP Morgan, and The Alfred P. Sloan Foundation.

## 7 REPRODUCIBILITY STATEMENT

For all the theorems, corollaries, and algorithms, we provide proofs and correctness analysis in App. E. To implement the algorithms, we provide pseudo-code in the main text and App. C. We also provide experiment specifications, environment setup, and neural network hyperparameters in App. F. Colored Sokoban and Button Maze are implemented based on Sokoban (Schrader, 2018) and GridWorld (Chevalier-Boisvert et al., 2018), respectively.

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

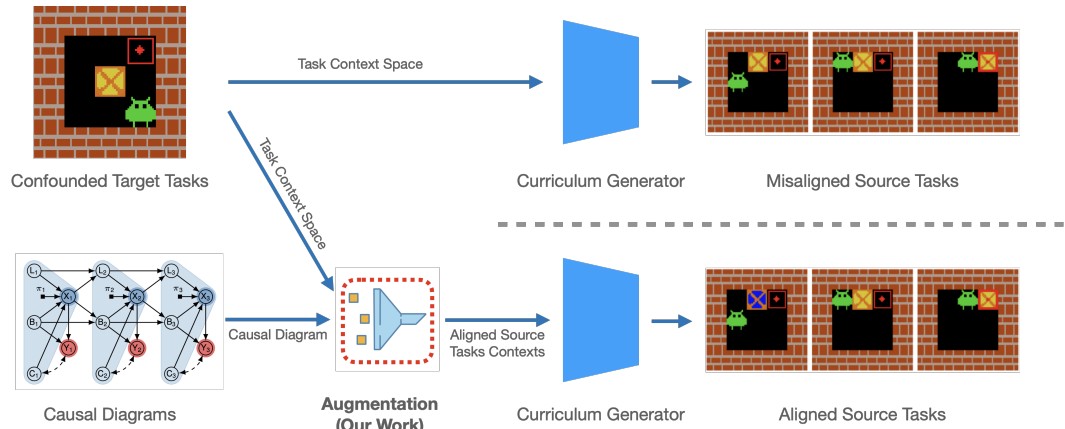

Figure 7: An overview of our proposed approach. In the upper half, the non-causal curriculum generator will produce misaligned source tasks failing to train the agent to convergence. While in our approach (the bottom half), we utilize the qualitative causal knowledge (causal diagrams) to filter out those misaligned source task contexts such that applying the same curriculum generator can now produce aligned source tasks.

Table 1: Time complexity of FINDMAXEDIT and its variations.

| Algorithm | Description | Runtime |
|---|---|---|
| ISEDIT | For given set of $\tau^{(j)}$, $\boldsymbol{X}^{(j)}$ and graph $\mathcal{G}_\pi$ decide editability | $\mathcal{O}(d(n+m))$ |
| FINDEDIT | Find an admissible set of $\tau^{(j)}$ w.r.t $\boldsymbol{X}^{(j)}$ in $\mathcal{G}_\pi$ | $\mathcal{O}(dn^2)$ |
| FINDMAXEDIT | Find the maximal admissible set of $\tau^{(j)}$ w.r.t $\boldsymbol{X}^{(j)}$ in $\mathcal{G}_\pi$ | $\mathcal{O}(dn^2)$ |
| LISTEDIT | List all admissible sets of $\tau^{(j)}$ w.r.t $\boldsymbol{X}^{(j)}$ in $\mathcal{G}_\pi$ | $\mathcal{O}(n)$ delay |

## A  METHOD OVERVIEW

In this section, we will summarize our proposed approach from a high-level view and provide a figure for better illustration of the pipeline.

## B  DETAILED COMPUTATIONS OF EXAMPLE 1

For the reward of pushing the box, we have,

$$\mathbb{E}\left[Y_i\middle|B_i = \text{"next to goal"}, C_i; \text{do}\left(X_i = \text{"push"}\right)\right] \tag{9}$$

$$= \mathbb{E}\left[Y_i\middle|B_i = \text{"next to goal"}, U_i; \text{do}\left(X_i = \text{"push"}\right)\right] \tag{10}$$

$$= 10 \times \mathbb{1}(U_i = 0) - 10 \times \mathbb{1}(U_i = 1) \tag{11}$$

$$= \begin{cases} 10 & \text{if } C_i = 0 \\ -10 & \text{if } C_i = 1 \end{cases} \tag{12}$$

After fixing the box color, we have,

$$\mathbb{E}\left[Y_i\middle|B_i = \text{"next to goal"}; \text{do}\left(C_i, X_i = \text{"push"}\right)\right] \tag{13}$$

$$= 10 \times P(U_i = 0) - 10 \times P(U_i = 1) \tag{14}$$

$$= -5. \tag{15}$$

---

**Algorithm 4:** ISEDIT

---

**Input:** A set of edit indicators $\tau^{(j)}$, a set of actions $\boldsymbol{X}^{(j)}$ and a graph $\mathcal{G}_\pi$.
**Output:** Whether the input edits are admissible.
**for** $X$ *in* $\boldsymbol{X}^{(j)}$ **do**
    **if** $\left(\tau^{(j)} \perp\!\!\!\perp \boldsymbol{Y} \cap De(X) \mid X, \boldsymbol{S}_X\right)$ *in* $\mathcal{G}_\pi$ **then**
        **return** False;
**return** True;

---

---

**Algorithm 5:** LISTEDIT

---

**Input:** A causal diagram $\mathcal{G}_\pi$, a set of actions $\boldsymbol{X}^{(j)}$.
**Output:** All possible editable sets.
$\Delta^{(j)} \rightarrow$FINDMAXEDIT$(\mathcal{G}_\pi, \boldsymbol{X}^{(j)})$;
mask $\leftarrow 2^{|\Delta^{(j)}|} - 1$;
**while** *mask* **do**
    $\Delta \leftarrow \emptyset$;
    tmpmask $\leftarrow$ mask;
    $i \leftarrow |\Delta^{(j)}| - 1$;
    **while** *tmpmask* **do**
        **if** *tmpmask* &1 **then**
            $\Delta \leftarrow \Delta \cup \{\Delta^{(j)}[i]\}$
        $i \leftarrow i - 1$;
        tmpmask $\gg 1$;
    mask$\leftarrow$ mask $-1$;
    **yield** $\Delta$;

---

## C VARIATIONS OF ALGORITHMS

The first task of deciding editability is a direct application of the criterion in Def. 4 and can be implemented by using TESTSEP (Algo. 4). Finding an editable set and finding the maximal editable set are both solved by our proposed Algo.1. After we find the maximal editable set, by Thm. 2, the power set of it is indeed all the possible editable sets w.r.t $\boldsymbol{X}^{(j)}$ in $\mathcal{G}_\pi$ (Algo. 5). Note that there are exponentially many subsets so we focus on the time complexity of yielding each output. We implement this by bit manipulation. Specifically, we denote the existence of a variable in the output set as 0 or 1. Then listing all the elements in a power set of $\Delta^{(j)}$ is equivalent to decrease a bit mask from $2^{|\Delta^{(j)}|} - 1$ to 0. Each mask in this process represents a unique subset of $\Delta^{(j)}$.

## D ALGORITHM FOR CAUSALLY ALIGNED CURRICULUM LEARNING

Algo. 3 construct the curriculum before any training is done. However, a more common practice is to construct the curriculum while adapting to the edge of the agent's capability. In this section, we will discuss a more involved version of Algo. 3 that implements this idea. As we have seen in the main text, two source tasks $\mathcal{T}^{(i)}, \mathcal{T}^{(j)}$ in a soluble curriculum satisfy $\boldsymbol{X}^{(i)} \subseteq \boldsymbol{X}^{(j)}$ if $i < j$. This incremental nature of the action sets actually brings us opportunities to reduce further the computation of finding editable states. Thm. 4 reveals a nice property of $\boldsymbol{X}^{(j)}$ such that if a set of state variables, $\Delta^{(j)}$, is editable w.r.t an action $X$, it is also editable w.r.t any actions $X'$ precedent to $X$ in the soluble ordering, $X' \prec X$, i.e., $(X' \cap De(Y) \perp\!\!\!\perp \pi \mid X', \boldsymbol{S}_{X'})$ in $\mathcal{G}_\pi$ where $\pi$ is a new regime node pointing to $X$.

**Theorem 4** (Expanded Action Sets). *Let* $\mathcal{T} = \langle \mathcal{M}, \Pi, \mathcal{R} \rangle$ *be a soluble target task and* $\mathcal{T}^{(j)} = \langle \mathcal{M}, \Pi, \mathcal{R}, \Delta^{(j)} \rangle$ *an aligned source task of* $\mathcal{T}$. *Then* $\Delta^{(j)}$ *is also editable w.r.t* $\boldsymbol{X}^{(j)+} = \left\{ X' \mid X' \prec X, X \in \boldsymbol{X}^{(j)} \right\}$.

---

**Algorithm 6:** CAUSAL CURRICULUM LEARNING

---

**Input:** A target task $\mathcal{T}$, a curriculum generator GEN.
**Output:** A policy trained in the curriculum $\mathcal{C}$, $\pi^*(\mathcal{C})$.
Generate causal diagram $\mathcal{G}_\pi$ of $\mathcal{T}$ w.r.t policy space $\Pi$;
Sort the set of actions w.r.t the soluble ordering (ascending) and save the result into $\boldsymbol{X}'$;
Randomly initialize the agent's policy $\pi^*(\mathcal{C})$;
$j \leftarrow 1$;
**for** $X_i$ *in* $\boldsymbol{X}'$ **do**
    $\Delta \leftarrow$ FINDMAXEDIT$(\mathcal{G}_\pi, \{X_i\})$;
    $\mathcal{C}[X_i] \leftarrow \emptyset$;
    **while** $\Omega(\boldsymbol{S}_i; \pi^*) \nsubseteq \bigcup_{\mathcal{T}^{(j)} \in \mathcal{C}[X_i]} \Omega(\boldsymbol{S}_i; \pi^*(\mathcal{C}))$ **do**
        Generate source task $\mathcal{T}^{(j)}$ with GEN$(\mathcal{T}, \Delta)$;
        Train $\pi^*(\mathcal{C})$ in $\mathcal{T}^{(j)}$ to converge;
        $\mathcal{C}[X_i] \leftarrow \mathcal{C}[X_i] \bigcup \{\mathcal{T}^{(j)}\}$;
        $j \leftarrow j + 1$;
**return** $\pi^*(\mathcal{C})$;

---

Thm. 4 implies that when checking editability w.r.t a set of actions, we only need to check w.r.t the single action that ranked highest in the soluble ordering (e.g., for actions $X_3 \prec X_2 \prec X_1$, finding the editable states for $\{X_1\}$ is equivalent to finding editable states for $\{X_i\}_{i=1}^3$).

Based on the above discussion and Algo. 1, we propose a causal curriculum learning algorithm (Algo. 6) to find the optimal target task policy given a curriculum generator GEN that generates source tasks by editing a given set of editable states $\Delta^{(j)}$. This algorithm resonates with the heuristics that the agent should be trained in a sequence of more challenging source tasks (Portelas et al., 2020; Narvekar et al., 2020). More specifically, after sorting the actions in the soluble ordering, the first generated source task only requires the agent to learn the optimal target policy of action $X_N$ (Say the target task $\mathcal{T}$ is an $N$-step one). The next stage of learning will be to grasp the optimal target policy of both action $X_N$ and $X_{N-1}$, but since the agent has already learned the optimal policy of $X_N$ previously. This won't be too much harder for it. We will continue this procedure until the agent learns the optimal policy for all the actions in the target task. Algo. 6 is basically a combination of CURRICULUM LEARNING(Algo. 2) and FINDCAUSALCURRICU-LUM(Algo. 3) except that for each $\Delta$, there will be multiple source tasks $\mathcal{T}^{(j)}$ being generated until $\Omega(\boldsymbol{S}_i; \pi^*) \subseteq \bigcup_{\mathcal{T}^{(j)} \in \mathcal{C}[X_i]} \Omega(\boldsymbol{S}_i; \pi^*(\mathcal{C}))$ where $\Omega(\boldsymbol{S}_i; \pi^*)$ is the space of possible $\boldsymbol{S}_i$ values in the target task under the optimal target task policy $\pi^*$, $\Omega(\boldsymbol{S}_i; \pi^*(\mathcal{C}))$ is the space of possible $\boldsymbol{S}_i$ values in source tasks under policy $\pi^*(\mathcal{C})$ and $\mathcal{C}[X_i]$ is the set of source tasks $\mathcal{T}^{(j)}$ such that $X_i \in \boldsymbol{X}^{(j)}$. This condition ensures that the input space of an action $X_i$ is thoroughly traversed during training in the curriculum. If this is satisfied, it means that the agent has learned the optimal decision rule for $X_i$ in every possible situation in the target task, as we show formally in the following theorem.

**Theorem 5.** *If the set of actions in target task satisfies $\boldsymbol{X} \subseteq \boldsymbol{X}^{(N)}$ and $\forall X_i \in \boldsymbol{X}$, $\Omega(\boldsymbol{S}_i; \pi^*) \subseteq \bigcup_{\mathcal{T}^{(j)} \in \mathcal{C}[X_i]} \Omega(\boldsymbol{S}_i; \pi^*(\mathcal{C}))$, the following solution $\pi^*(\mathcal{C})$ is an optimal policy $\pi^*$ in the target task $\mathcal{T}$,*

$$\pi^*(\mathcal{C}) = \arg\max_{\pi \in \Pi} \mathbb{E}_{\mathcal{M}^{(N)}} \left[ \mathcal{R}(\boldsymbol{Y}); \pi \right], \tag{16}$$

*where $\mathcal{M}^{(N)}$ is the SCM of source task $\mathcal{T}^{(N)}$, and $\mathcal{T}^{(N)}$ is aligned to $\mathcal{T}$ w.r.t $\boldsymbol{X}^{(N)}$.*

Nonetheless, there are other practical roadblocks before realizing optimal curriculum learning. For example, we don't have an exact measurement of when the input space of an action is traversed thoroughly by the curriculum, and the agent may not converge in every source task $T^{(j)}$ to learn all the optimal decision rules of $\boldsymbol{X}^{(j)}$. But still, as we have shown in the experiments, simply augmenting the existing curriculum generators already gives us satisfying performance.

## E PROOF FOR THEOREMS AND ALGORITHM CORRECTNESS

In this section, we provide proof sketches for all the theorems proposed in the previous sections.

**Lemma 1.** *If the set of edit indicators satisfy $(\tau^{(j)} \perp\!\!\!\perp \boldsymbol{Y}_X | X, \boldsymbol{S}_X)$ in a soluble source task, for any $X' \prec X$ in the soluble ordering, it satisfies $(\tau^{(j)} \perp\!\!\!\perp \boldsymbol{Y}_{X'} | X', \boldsymbol{S}_{X'})$.*

*Proof of Lem. 1.* We will prove this by contradiction. Suppose $(\tau^{(j)} \not\perp\!\!\!\perp \boldsymbol{Y}_{X'} | X', \boldsymbol{S}_{X'})$. There are three possibilities based on the type of node $V$ to which $\tau^{(j)}$ corresponds.

i) If $V \in \boldsymbol{S}_X$, we can show that the source task cannot be soluble by finding an open path from a pseudo parent of $X$ to $\boldsymbol{Y}_{X'}$. Firstly, $X' \prec X$ in a soluble source task means $X' \in De(X)$. When adding a pseudo parent $V'$ to $X$, the path $V' \rightarrow X \leftarrow V \leftarrow \tau^{(j)}$ is open under $X', \boldsymbol{S}_{X'}$. By the assumption, there also exists an open path from $\tau^{(j)}$ to $\boldsymbol{Y}_{X'}$ under $X', \boldsymbol{S}_{X'}$, which means that path $V' \rightarrow X \leftarrow V \leftarrow \tau^{(j)}$ is also open under $X', \boldsymbol{S}_{X'}$. Thus, this contradicts the definition of a soluble task and $X' \prec X$.

ii) If $V \notin \boldsymbol{S}_X$ and $V \in An(X)$, we can find a similar open path as in the previous case. By the assumption, there exists an open path $p$ from $\tau^{(j)}$ to $\boldsymbol{Y}_{X'}$ under $X', \boldsymbol{S}_{X'}$. But we have $X' \in De(X)$ and thus $\boldsymbol{Y}_{X'} \subseteq \boldsymbol{Y}_X$. This open path must be blocked under $X, \boldsymbol{S}_X$, which means that $X$ must observe a non-collider variable $Z$ on $p$. Without observing $Z$, $p$ will be open under $X, \boldsymbol{S}_X$. Note that any colliders on $p$ will have $X'$ as their descendant since $p$ is open under $X', \boldsymbol{S}_{X'}$, which creates new open paths. So, $p$ won't be blocked because of colliders. Now, a pseudo parent of $X$, say $V'$, will again have an open path towards $\boldsymbol{Y}_{X'}$ under $X', \boldsymbol{S}_{X'}$, $V' \rightarrow X \leftarrow Z \cdots \rightarrow \boldsymbol{Y}_{X'}$, where $Z \cdots \rightarrow \boldsymbol{Y}_{X'}$ is part of the $p$ path. Thus, this contradicts the definition of a soluble task and $X' \prec X$.

iii) If $V \notin An(X)$, we prove contradictions by checking all the possible path types between $V$ and $\boldsymbol{Y}_{X'}$. If the open path $p$ from $V$ to $\boldsymbol{Y}_{X'}$ under $X', \boldsymbol{S}_{X'}$ is causal, $p$ must be blocked under $X, \boldsymbol{S}_X$ by having $X$ observe variables on $p$, which contradicts with the condition that $V \notin An(X)$. If $p$ is not causal and colliders exist, say $Z$, $X'$ must be a descendant of such colliders. Let $Z$ be the leftmost collider on $p$. This leads to another causal path $p'$, $\tau^{(j)} \rightarrow V \rightarrow \cdots \rightarrow Z \rightarrow \cdots \rightarrow X' \rightarrow \cdots \rightarrow \boldsymbol{Y}_{X'}$. Clearly, $X$ cannot block $p'$ by observing any variables on it. Thus, $p'$ is open under $X, \boldsymbol{S}_X$ which contradicts with the condition that $(\tau^{(j)} \perp\!\!\!\perp \boldsymbol{Y}_X | X, \boldsymbol{S}_X)$.

Thus, we have proved that if $(\tau^{(j)} \perp\!\!\!\perp \boldsymbol{Y}_X | X, \boldsymbol{S}_X)$ in a soluble source task, for any $X' \prec X$, it also satisfies $(\tau^{(j)} \perp\!\!\!\perp \boldsymbol{Y}_{X'} | X', \boldsymbol{S}_{X'})$. $\qquad \square$

*Proof of Thm. 1.* By definition, the optimal policy of the target task satisfies,

$$\pi^* = \arg\max_{\pi \in \Pi} \mathbb{E}_{\mathcal{M}} \left[ \mathcal{R}(\boldsymbol{Y}); \pi \right]. \tag{17}$$

Similar definitions hold for the optimal policy $\pi^{(j)}$ of a source task $\mathcal{T}^{(j)}$. Our goal is to show that once the graphical criterion holds for $\tau^{(j)}$ and action $X_i$, the optimal decision rule of $X_i$ on those shared input states of the source task and the target task is the same in these two tasks. We first simplify the calculation of the optimal decision rule at action $X_i$ using the concept of "relevance graph"(Koller & Milch, 2003).

**Definition 8** (Relevance Graph of Tasks). The relevance graph, $\mathcal{G}_r$, of a target task $\mathcal{T} = \langle \mathcal{M}, \Pi, \mathcal{R} \rangle$ is a directed graph whose nodes representing action variables $\boldsymbol{X} \subseteq \boldsymbol{V}$ are connected by directed edges $X' \rightarrow X$ if and only if $(\pi' \not\perp\!\!\!\perp \boldsymbol{Y} \cap De(\boldsymbol{X}) | \boldsymbol{S}_X, X)$ where $\pi'$ is an added regime node pointing to $X'$.

This relevance graph specifies the order in which those actions should be optimized. Intuitively, $X'$ is optimized before $X$ because it can affect $X$'s reward signal while $X$'s inputs, $\boldsymbol{S}_X$, and $X$ itself cannot block such causal effects. We denote the topological order over actions in $\mathcal{G}_r$ by $\prec$ where $X' \prec X$ if and only if $X'$ should be optimized before $X$. It is also possible that the relevance graph contains Strongly Connected Components (SCCs) where each pair of actions in the same SCC is connected by a directed path. Semantically, actions in the same SCC affect the rewards of each other. Thus, one must consider all actions in the same SCC to find the optimal policy for these actions (Koller & Milch, 2003). Formally speaking, let $\mathcal{L}$ be an equivalence relationship over $\boldsymbol{X}$ such that action $X, X' \in \boldsymbol{X}$ belong to the same partition, $X \sim_{\mathcal{L}} X'$, (equivalently, $X \in [X']_{\mathcal{L}}$ or $X' \in [X]_{\mathcal{L}}$) if and only if they are in the same maximal SCC of a target task $\mathcal{T}$'s relevance

graph, $\mathcal{G}_r$.[4] When there is no SCC in the relevance graph, we say this target task is soluble Def. 7, which can be solved by a series of dynamic programs (Koller & Milch, 2003; Lauritzen & Nilsson, 2001). When there is an SCC, we can treat each SCC as a single high-level action taking value as the combination of all the actions in that SCC, i.e., $\Omega([X_i]) = \times_{X \in [X_i]} \Omega(X)$. After eliminating all such SCCs in the relevance graph, we transform the task back to a soluble one. So, the same solver can be used. Now we show that given an optimal target task policy, the decision rule at $[X_i]$ is optimal if and only if for every input $\boldsymbol{s}_{[X_i]} \in \Omega(\boldsymbol{S}_{[X_i]}; \pi^*)$,

$$\pi^*_{[X_i]}(\cdot|\boldsymbol{s}_{[X_i]}) = \underset{\pi_{[X_i]}(\cdot|\boldsymbol{s}_{[X_i]})}{\arg\max} \; \mathbb{E}_\mathcal{M} \left[ \mathcal{R}(\boldsymbol{Y}_{[X_i]}) \mid \boldsymbol{s}_{[X_i]}; \pi_{[X_i]} \cup \pi^*_{\prec[X_i]} \right], \tag{18}$$

where $\boldsymbol{Y}_{[X_i]} = De([X_i]) \cap \boldsymbol{Y}$ is the set of rewards that are descendants of actions in the SCC of $X_i$ and $\pi^*_{\preceq X_i}$ is the set of optimal decision rules of actions that precede $[X_i]$ w.r.t the relevance graph. By definition, the decision rule at $[X_i]$ is optimal if it maximizes the expected reward function when other parts of the optimal policy are given,

$$\pi^*_{[X_i]} = \underset{\pi_{[X_i]}}{\arg\max} \mathbb{E}_\mathcal{M} \left[ \mathcal{R}(\boldsymbol{Y}); \pi_{[X_i]} \cup (\pi^* \setminus \pi^*_{[X_i]}) \right]. \tag{19}$$

We can expand the expected reward as follows,

$$\mathbb{E}_\mathcal{M} \left[ \mathcal{R}(\boldsymbol{Y}); \pi_{[X_i]} \cup (\pi^* \setminus \pi^*_{[X_i]}) \right] = \sum_{\boldsymbol{s}_{[X_i]}} P(\boldsymbol{s}_{[X_i]}) \mathbb{E}_\mathcal{M} \left[ \mathcal{R}(\boldsymbol{Y}) \mid \boldsymbol{s}_{[X_i]}; \pi_{[X_i]} \cup (\pi^* \setminus \pi^*_{[X_i]}) \right]. \tag{20}$$

Clearly, the input distribution of $\boldsymbol{S}_{[X_i]}$ is fixed given other parts of the optimal policy so Eq. (19) is equivalent to for every input $\boldsymbol{s}_{[X_i]} \in \Omega(\boldsymbol{S}_{[X_i]}; \pi^*)$,

$$\pi^*_{[X_i]}(\cdot|\boldsymbol{s}_{[X_i]}) = \underset{\pi_{[X_i]}(\cdot|\boldsymbol{s}_{[X_i]})}{\arg\max} \; \mathbb{E}_\mathcal{M} \left[ \mathcal{R}(\boldsymbol{Y}) \mid \boldsymbol{s}_{[X_i]}; \pi_{[X_i]} \cup (\pi^* \setminus \pi^*_{[X_i]}) \right] \tag{21}$$

$$= \underset{\pi_{[X_i]}(\cdot|\boldsymbol{s}_{[X_i]})}{\arg\max} \sum_{\boldsymbol{y}, [x_i]} \mathcal{R}(\boldsymbol{y}) P(\boldsymbol{y}|\boldsymbol{s}_{[X_i]}, [x_i]; \pi^* \setminus \pi^*_{[X_i]}) \pi([x_i]|\boldsymbol{s}_{[X_i]}). \tag{22}$$

Since non-descendants of $[X_i]$ are independent of $[X_i]$ given $\boldsymbol{S}_{[X_i]}$ and actions are not confounded, we can further reduce the reward function to focus on only $\boldsymbol{Y}_{[X_i]}$, rewards that are descendants of $[X_i]$, assuming that $\mathcal{R}(\cdot)$ is cumulative,

$$\pi^*_{[X_i]}(\cdot|\boldsymbol{s}_{[X_i]}) = \underset{\pi_{[X_i]}(\cdot|\boldsymbol{s}_{[X_i]})}{\arg\max} \sum_{\boldsymbol{y}_{[X_i]}, [x_i]} \mathcal{R}(\boldsymbol{y}_{[X_i]}) P(\boldsymbol{y}_{[X_i]}|\boldsymbol{s}_{[X_i]}, [x_i]; \pi^* \setminus \pi^*_{[X_i]}) \pi([x_i]|\boldsymbol{s}_{[X_i]}). \tag{23}$$

From the relevance graph definition, we know that only actions that precede $[X_i]$ in the relevance graph will affect $\boldsymbol{Y}_{[X_i]}$ given $\boldsymbol{S}_{[X_i]}, [X_i]$. Thus, we can simplify the conditioning further,

$$\pi^*_{[X_i]}(\cdot|\boldsymbol{s}_{[X_i]}) = \underset{\pi_{[X_i]}(\cdot|\boldsymbol{s}_{[X_i]})}{\arg\max} \sum_{\boldsymbol{y}_{[X_i]}, [x_i]} \mathcal{R}(\boldsymbol{y}_{[X_i]}) P(\boldsymbol{y}_{[X_i]}|\boldsymbol{s}_{[X_i]}, [x_i]; \pi^*_{\prec[X_i]}) \pi([x_i]|\boldsymbol{s}_{[X_i]}) \tag{24}$$

$$= \underset{\pi_{[X_i]}(\cdot|\boldsymbol{s}_{[X_i]})}{\arg\max} \; \mathbb{E}_\mathcal{M} \left[ \mathcal{R}(\boldsymbol{Y}_{[X_i]}) \mid \boldsymbol{s}_{[X_i]}; \pi_{[X_i]} \cup \pi^*_{\prec[X_i]} \right], \tag{25}$$

which is exactly Eq. (18). We can do a similar derivation for the optimal decision rule at $[X_i]$ in source task $\mathcal{T}^{(j)}$, for every input $\boldsymbol{s}_{[X_i]} \in \Omega^{(j)}(\boldsymbol{S}_{[X_i]}; \pi^{(j)})$,

$$\pi^{(j)}_{[X_i]}(\cdot|\boldsymbol{s}_{[X_i]}) = \underset{\pi_{[X_i]}(\cdot|\boldsymbol{s}_{[X_i]})}{\arg\max} \sum_{\boldsymbol{y}_{[X_i]}, [x_i]} \mathcal{R}(\boldsymbol{y}_{[X_i]}) P(\boldsymbol{y}_{[X_i]}|\boldsymbol{s}_{[X_i]}, [x_i], \tau^{(j)}; \pi^{(j)}_{\prec[X_i]}) \pi([x_i]|\boldsymbol{s}_{[X_i]}) \tag{26}$$

$$= \underset{\pi_{[X_i]}(\cdot|\boldsymbol{s}_{[X_i]})}{\arg\max} \; \mathbb{E}_{\mathcal{M}^{(j)}} \left[ \mathcal{R}(\boldsymbol{Y}_{[X_i]}) \mid \boldsymbol{s}_{[X_i]}; \pi_{[X_i]} \cup \pi^{(j)}_{\prec[X_i]} \right], \tag{27}$$

where $\tau^{(j)}$ is the edit indicator $\pi^{(j)}$ is the optimal policy of source task $\mathcal{T}^{(j)}$. In practice, we can let actions in each SCC $X_i$ have an edge into every reward node associated with the SCC and still

---

[4]for simplicity, we will use $[X]$ to denote $[X]_\mathcal{L}$ in the following discussion.

the graph is compatible with the original task. Then, we only need to show that once the graphical criterion is satisfied in such a graph, for any input $s_i \in \Omega^{(j)}(S_i; \pi^{(j)}) \cap \Omega(S_i; \pi^*)$, the following holds,

$$P(y_{X_i}|s_{[X_i]}, [x_i]; \pi^*_{\prec[X_i]}) = P(y_{X_i}|s_{[X_i]}, [x_i], \tau^{(j)}; \pi^{(j)}_{\prec[X_i]}). \tag{28}$$

If this is true, the optimal decision rule at $X_i$ will be invariant across both the target task and the source task. We first consider the simpler case when there is no SCC in the relevance graph. So, $[X_i] = \{X_i\}$. In this case, we can apply the result of Lem. 1 directly and know that the graphical criterion is also satisfied by any action $X \prec X_i$. Then we prove Eq. (28) holds by induction on action $X_i$. The base case is there is no action preceding $X_i$ in the relevance graph. So, there will be no policy dependencies in Eq. (28), and it will be trivially true given the graphical criterion. Now we assume when there are $k$ actions preceding $X_i$ in the relevance graph, and the graphical criterion holds, Eq. (28) will hold. When there are $k + 1$ actions preceding $X_i$ in the relevance graph, by the inductive hypothesis and the fact that graphical criterion also holds for $X \prec X_i$ when it holds for $X_i$, we know the optimal decision rules of $X \prec X_i$ stay the same across the source task and the target task. Thus, Eq. (28) is reduced to $P(y_{X_i}|s_{X_i}, x_i) = P(y_{X_i}|s_{X_i}, x_i, \tau^{(j)})$, which is true when the graphical criterion holds, $(\tau^{(j)} \perp\!\!\!\perp Y_{X_i} \mid X_i, S_i)$ in $\mathcal{G}_\pi$.

When there are SCCs in the relevance graph, the graphical criterion only shows us that $(\tau^{(j)} \perp\!\!\!\perp Y_{X_i} \mid S_i, X_i)$ in $\mathcal{G}_\pi$. But we can show that $(\tau^{(j)} \perp\!\!\!\perp Y_{X_i} \mid [X_i], S_{[X_i]})$ in $\mathcal{G}_\pi$ also holds. Notice that the difference between these two criteria is that the latter includes more variables in the conditioning set. If the latter one doesn't hold, that means adding these variables opens up at least a collider path $p$ that is blocked under the original criterion. Say this collider on $p$ is $M$, and it's ancestral to an action $X_j \in [X_i]$. Clearly, there is an active path from $\tau^{(j)}$ to $M$ and there is also an active causal path from $M$ to $Y_{X_j}$ under $S_i, X_i$. But we also know that by the condition that $X_j \in [X_i]$, $Y_{X_j} \cap Y_{X_i}$. This means that there is also an active path from $\tau^{(j)}$ to $Y_{X_i}$ under $S_i, X_i$ which clearly contradicts with the fact that $(\tau^{(j)} \perp\!\!\!\perp Y_{X_i} \mid X_i, S_i)$ in $\mathcal{G}_\pi$. Thus, $(\tau^{(j)} \perp\!\!\!\perp Y_{X_i} \mid [X_i], S_{[X_i]})$ in $\mathcal{G}_\pi$ holds. Now, we can view the whole $[X_i]$ as one high-level action and follow a similar induction procedure as in the simpler case, which completes the proof.

$\square$

*Proof of Thm. 2.* We first show that the maximal editable set w.r.t. a set of actions is unique. Let $K = \max_{V_I^{(j)}} |V_I^{(j)}|$. If there are two maximal admissible sets $V_1, V_2$ w.r.t $X(j)$, they satisfy $|V_1| = |V_2| = K$ but $V_1 \neq V_2$. Interchangeably, we can assume a state variable $V \in V_1$ but $V \notin V_2$ exists. Since $V$ satisfies our criterion, so does every variable in $V_2$. Then by Def. 4, we know the set $V' = V_2 \cup \{V\}$ is admissible w.r.t $X^{(j)}$, which contradicts with the fact that $V_2$ is the maximal set since $|V'| = K + 1$. Thus, this is impossible to happen.

By the uniqueness, we only need to search for one maximal editable set w.r.t a given set of actions $X^{(j)}$. For each variable, by Def. 4, it has to be admissible w.r.t $X^{(j)}$ before it can be added to the admissible set $\Delta^{(j)}$. Thus, we loop through all state variables and check their editability. If a single state variable is not admissible w.r.t an action, $X \in X^{(j)}$, we don't need to check its editability w.r.t other actions further. So, we can break the loop there. The editability check is done on the augmented graph $\mathcal{G}_\pi$ where a pseudo edit indicator $\tau$ is added, pointing to $V$, the state variable being checked. The correctness of this step is guaranteed by the correctness of TESTSEP (van der Zander et al., 2014).

$\square$

*Proof of Thm. 3.* Since every source task we use is causally aligned, the optimal decision rules of actions in $X^{(j)}$ will be invariant across the target task and the source task $\mathcal{T}^{(j)}$. The same is true for the set of actions $X^{(j+1)}$. By our construction, we have $X^{(j)} \subseteq X^{(j+1)}$, and each action corresponds to exactly one element in $\pi^{(j)} \cap \pi^*, \pi^{(j+1)} \cap \pi^*$, respectively. Thus, the set of invariant optimal decision rules is also expanding.

$\square$

*Proof of Corol. 1.* By the correctness of FINDMAXEDIT, we know that every source task generated by GEN($\mathcal{T}, \Delta^{(j)}$) will be causally aligned w.r.t $X^{(j)}$. Then by the way we construct $X^{(j)}$, it is

guaranteed that $\boldsymbol{X}^{(j)} \subseteq \boldsymbol{X}^{(j+1)}$. Thus, the returned curriculum of FINDCAUSALCURRICULUM will be causally aligned. $\square$

*Proof of Thm. 4.* This is a direct result of applying Lem. 1 to soluble target tasks. $\square$

*Proof of Thm. 5.* The algorithm works by creating source tasks with an expanding set of actions. The set of actions expands in the direction that follows the soluble ordering, $\boldsymbol{X}' = \{X_N, X_{N-1}, ..., X_1\}$. Then, by Thm. 4, the editable set can be calculated w.r.t only the newly added action in this round ($X_i$). The while loop ensures we generate enough source tasks to cover all the possible state inputs to $X_i$. Note that we don't require the final output $\pi^*(\mathcal{C})$ to be optimal in all source tasks. It is still the optimal target task policy. Because given the expanding action sets when constructing source tasks, optimal decision rules of $\boldsymbol{X}^{(j-1)}$ learned in $T^{(j-1)}$ are still optimal in $T^{(j)}$ and the final output policy $\pi^*(\mathcal{C})$ will contain optimal decision rules for all actions $\boldsymbol{X}$ in the target task. $\square$

# F  EXPERIMENT AND IMPLEMENTATION DETAILS

This section introduces the details of our experiments, including environment specifications, agent hyper-parameters, training/testing protocols, and more experimental results. For the Colored Sokoban and the Button Maze, we implemented a Proximal Policy Optimization (PPO) agent with independent actor and critic networks in PyTorch (Schulman et al., 2017). Both networks have three convolutional layers and two linear layers with rectified linear unit (ReLU) activation functions. The number of output channels of the convolutional layers is $[32, 64, 64]$ with each layer. For convolutional kernels in those three convolutional layers, we use $8 \times 8$ with a stride of 4, $4 \times 4$ with a stride of 2, and $3 \times 3$ with a stride of 1, respectively. We flatten the output of the convolutional layers and feed it into the two linear layers with the intermediate feature dimension set to $512$. The input for the network is an image observation of size $84 \times 84 \times 3$ for both environments. For hyper-parameters, we follow the default hyper-parameters from Huang et al. (2022b) on which the implementation is also heavily based. For the Continuous Button Maze environment, we use an Soft Actor Critic (SAC) agent with low-dimensional state vector inputs following the implementations by Huang et al. and adopting the hyper parameters setting from Klink et al.. For the four curriculum generators used, we adopted the implementation from Klink et al. (2022)'s official implementations (https://github.com/psclklnk/currot). Note that even though all three environments are confounded, we still use MDP-based policy learning algorithms. Because for those environments, in both target task and aligned source tasks, the confounder is revealed by other variables without intervention. Thus, the agent has perfect information to decide which state it is in exactly. So, we can still use PPO and SAC to find the optimal policy.

## F.1  ENVIRONMENT SPECIFICATIONS

For Colored Sokoban, the target task definition is already specified in Example 1.

For Button Maze, at each time step, let $C_i$ be the target location's color, $B_i$ be the button status of whether it has been pushed or not, $L_i$ be the agent's current location, $Y_i$ be the reward for this step and $X_i$ be the agent's action in this step. $B_i = \neg B_{i-1}$ when the agent pushes the button. The goal location's color $C_i = U_i$ before the button is pushed but $C_i = U_C$ after the button is pushed, where $P(U_i = 1) = 1/2, P(U_C = 1) = 1/5$. The reward function is specified as follows,

$$Y_i = \begin{cases} 1 & \text{if } B_i = \text{``next to goal''} \wedge X_i = \text{``move forward''} \wedge (U_C = 1) \\ -1 & \text{if } B_i = \text{``next to goal''} \wedge X_i = \text{``move forward''} \wedge (U_C = 0) \\ -0.1 & \text{if } L_i = L_{i-1} \\ -0.01 & \text{otherwise} \end{cases}. \tag{29}$$

In the Continuous Button Maze environment Fig. 8, similar to the grid-world button maze, the agent also must navigate to the target region at the right time. The only difference is that this time the environment is an open area and all states and actions are in the continuous domain, which is exponentially large. The optimal strategy for the agent is still to push the button first then step onto

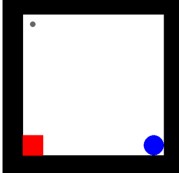

Figure 8: Continuous Button Maze. The agent (the grey dot) needs to navigate the room and step into the goal region (the region in the bottom left) at the right time. The goal region flashes between red and green. And after pushing the button, it will be always green.

the green goal region. Note that the goal region flashes between red and green color before pushing the button. Even if the agent step onto it when it shows green, it is still possible that the agent gets a negative reward. The environment is defined the same as the Button Maze with the only difference that we remove the not-moving penalty from the reward function,

$$
Y_i = \begin{cases} 1 & \text{if } B_i = \text{"next to goal"} \wedge X_i = \text{"move to the goal"} \wedge (U_C = 1) \\ -1 & \text{if } B_i = \text{"next to goal"} \wedge X_i = \text{"move to the goal"} \wedge (U_C = 0) \,. \\ -0.01 & \text{otherwise} \end{cases} \tag{30}
$$

The curriculum generators are allowed to pick the initial location of the agents, whether to toggle the button at the beginning, and whether to interven the goal region color.

## F.2 Additional Experiment Results

When reporting results, instead of using the cumulative rewards directly, as promoted in Agarwal et al. (2021), we report the Interquartile Mean (IQM) normalized by the maximum and minimum rewards in the corresponding environment. Specifically, for each experiment, we run five random seeds. Then, the normalized IQM is calculated as,

$$
\text{NormalizedIQM} = \frac{2}{n} \sum_{i=\frac{n}{4}+1}^{\frac{3n}{4}} \frac{(x_i - l)}{(u - l)} \tag{31}
$$

where $x_i$ are the original data points we collected and sorted, e.g., cumulative rewards, and $\forall x_i, x_i \in [b, u], b, u \in \mathbb{R}$ are the bounds for the data points. In our experiments, the bounds for the rewards are easily obtainable as they are both artificially designed game environments. In the implementation, we edit the environment by editing a set of predefined parameters. Each vector of parameters corresponds to a unique instance of the environment.

In Fig. 9, we see that agent trained by causal agnostic curriculum generators fail to avoid misaligned source tasks and unable to converge to the optimal policy. Those agents trained by the curriculum generators perform even worse than those directly trained in the target task which verifies the necessity of avoiding misaligned source tasks empirically. After augmenting the same curriculum generators with our algorithm, agents trained by them all successfully converge to the optimal surpassing those trained directly in the target task. The result demonstrates that our method is indeed widely applicable to both high-dimensional and continuous domains and that utilizing qualitative causal knowledge properly is crucial to the successful application of curriculum learning in the confounded environments.

During the training process, we also counted the portions of misaligned source tasks generated by those non-causal curriculum generators. From Table 2, we see clearly that in all three environments, those curriculum generators fail to avoid misaligned source tasks and most of the source tasks proposed by them are actually misaligned. We report those portions with $95\%$ confidence intervals.

We also show examples of the curricula generated by those augmented generators in Figs. 10 to 12.

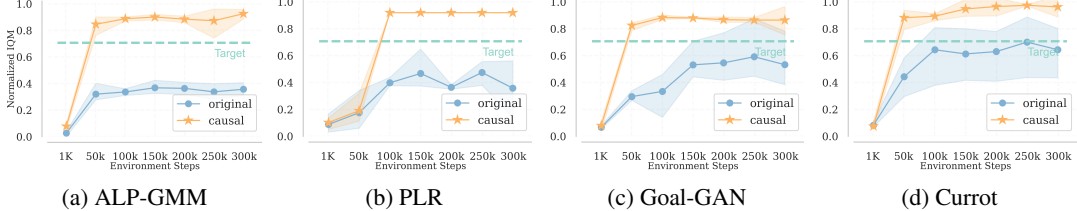

|  |  |  |  |
|---|---|---|---|
| (a) ALP-GMM | (b) PLR | (c) Goal-GAN | (d) Currot |

Figure 9: Target task performance of the agents at different training stages in the Continuous Button Maze using different curriculum generators (Columns). The horizontal green line shows the performance of the agent trained directly in the target. "original" refers to the unaugmented curriculum generator and "causal" refers to its causally augmented version.

Table 2: Misaligned Source Task Portion.

| Env./Alg. | ALP-GMM | PLR | Goal-GAN | Currot |
|---|---|---|---|---|
| **Colored Sokoban** | $68.10 \pm 0.71\%$ | $69.10 \pm 2.77\%$ | $66.41 \pm 1.84\%$ | $68.36 \pm 1.06\%$ |
| **Button Maze** | $88.41 \pm 0.43\%$ | $87.33 \pm 3.77\%$ | $90.30 \pm 0.27\%$ | $88.62 \pm 1.00\%$ |
| **Con. Button Maze** | $85.77 \pm 0.83\%$ | $83.57 \pm 14.5\%$ | $80.55 \pm 19.6\%$ | $47.70 \pm 32\%$ |

## G    DETAILED RELATED WORK

Research in generating suitable curricula for reinforcement learning agents can date back to the 90s when people manually designed subtasks for robot controlling problems Sanger (1994). In recent works, a general curriculum generation framework requires two components, an encoded task space and a task characterization function (Narvekar et al., 2020; Wang et al., 2020). Each component may be either hand-coded as inputs (Khan et al., 2011; Peng et al., 2018; MacAlpine & Stone, 2018; Portelas et al., 2019) or automatically learned from data along with training the agents (Parker-Holder et al., 2022; Klink et al., 2022; Florensa et al., 2018; Jiang et al., 2021; Florensa et al., 2017; Risi & Togelius, 2020; Cho et al., 2023; Huang et al., 2022a). A straightforward task space encoding is to split the observation with common patterns (e.g. pixel tiles) and re-combine them to create new tasks (Dahlskog & Togelius, 2014), which can be intractable in the face of a rich observation space and adds extra representational burdens to the curriculum generator. Recent work usually uses vectored parameters as task space encoding, each of which can be grounded into a unique task instance (Parker-Holder et al., 2022; Klink et al., 2022; Florensa et al., 2018; Jiang et al., 2021; Portelas et al., 2019; Dennis et al., 2020; Wang et al., 2019; 2020; Cho et al., 2023; Huang et al., 2022a). While parameter-based encoding is widely applicable in various decision-making tasks, using a dedicated domain description language such as Video Game Description Language (VGDL) for task space encoding provides finer granularity and readability in task generation (Schaul, 2013; Liebana et al., 2016; Justesen et al., 2018). A suitable task space encoding then lays the foundation of a reasonable task characterization function, which is either a task difficulty measure (Florensa et al., 2018; Parker-Holder et al., 2022; Dennis et al., 2020; Andreas et al., 2017; Sukhbaatar et al., 2018; Jiang et al., 2021) or a task similarity function (Svetlik et al., 2017; Silva & Costa, 2018; Jiang et al., 2021; Eysenbach et al., 2019) in general. For example, task similarity can be measured via domain knowledge based heuristics (Svetlik et al., 2017; Andrychowicz et al., 2017; Silva & Costa, 2018) and agent's performance can be used as a direct indicator of the task difficulty (Florensa et al., 2017; 2018; Narvekar et al., 2017; Parker-Holder et al., 2022). Curriculum generator relies heavily on these task characteristic functions to measure the quality of the task, schedule the training process, and evaluate the agent's performance (Narvekar et al., 2020; Portelas et al., 2020).

Given task space encoding and task characteristic functions, the remaining central problem of curriculum learning is how exactly one could generate new tasks efficiently. The most intuitive idea of training agents on increasingly harder tasks has been verified in various works, which can be implemented as setting different goals (Florensa et al., 2018; Racanière et al., 2019; Baranes & Oudeyer, 2013) or changing starting state distributions (Florensa et al., 2017; Salimans & Chen, 2018; Asada et al., 1996; Narvekar et al., 2016). Another major branch of task generation is to change task's state space or task parameters. This approach usually works together with a parametrized task space

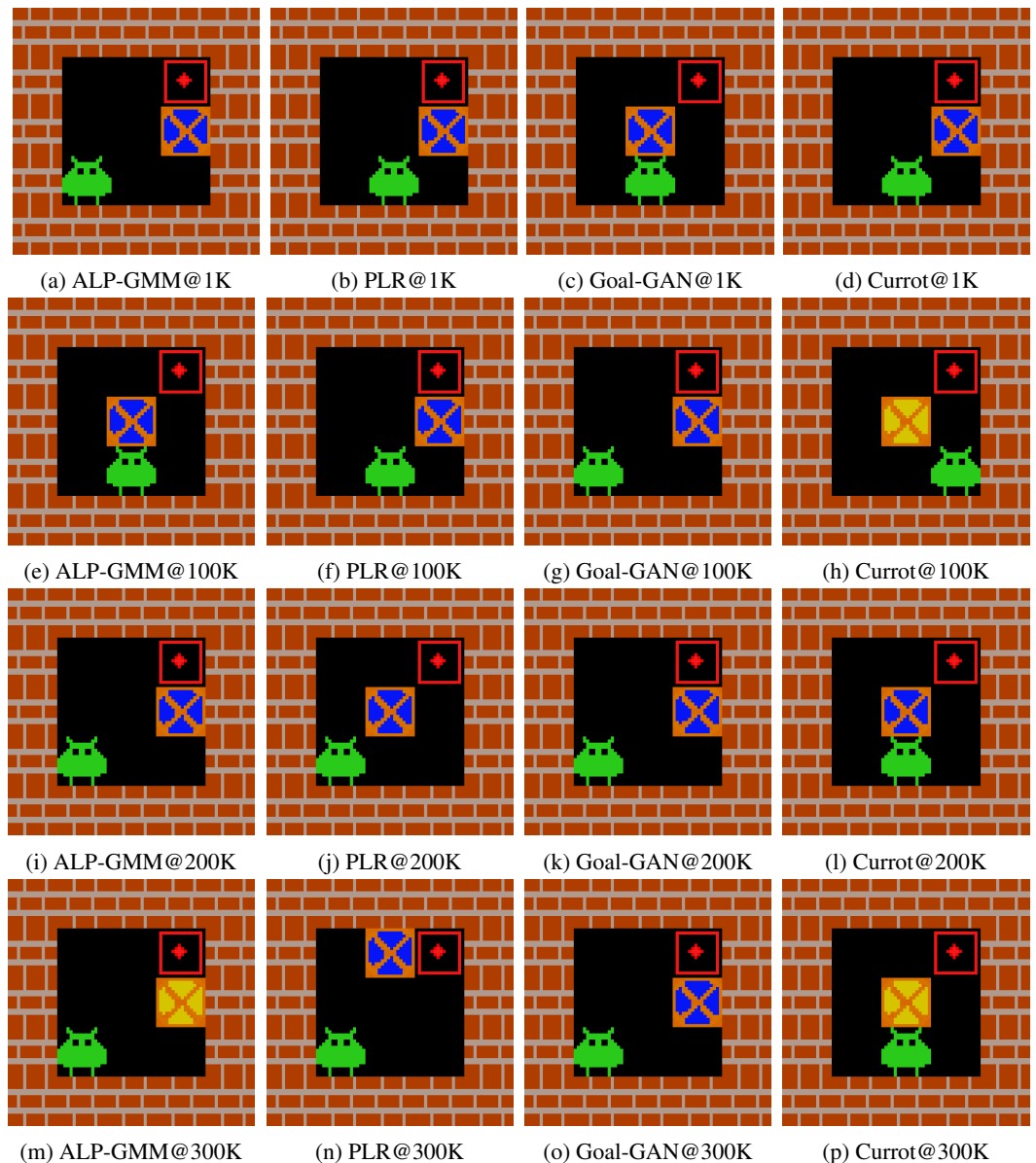

Figure 10: Curricula generated by the (causal) augmented curriculum generators for Colored Sokoban. Each column shows a curriculum from one generator at different training steps.

where all state factors and transition function parameters are fully encoded as a latent vector (Parker-Holder et al., 2022; Klink et al., 2022; Florensa et al., 2018; Jiang et al., 2021; Portelas et al., 2019; Dennis et al., 2020; Wang et al., 2019; 2020; Cho et al., 2023; Huang et al., 2022a). When more flexibility is desired, source tasks can also be generated by changing the transition or reward function forms. In curiosity-driven agents, exploration is encouraged by an intrinsic reward added upon the original reward from the task (Bellemare et al., 2016; Ecoffet et al., 2019), and this intrinsic reward signal can be learnable which evolves as agents progress in the task (Burda et al., 2019; Singh et al., 2004). Another telling illustration of function forms change is the Sim2Real problem. When generating tasks for a robot in simulation, other than a simulated perfect dynamics model, one needs to take into account extra fractions, inaccurate sensors, motor latency, and poorly executed actions for a successful deployment into the real-world target task (OpenAI et al., 2019).

The crux of curriculum reinforcement learning is to transfer useful knowledge from subtasks to target tasks (Narvekar et al., 2020), which can be modeled as a transportability problem in the causal

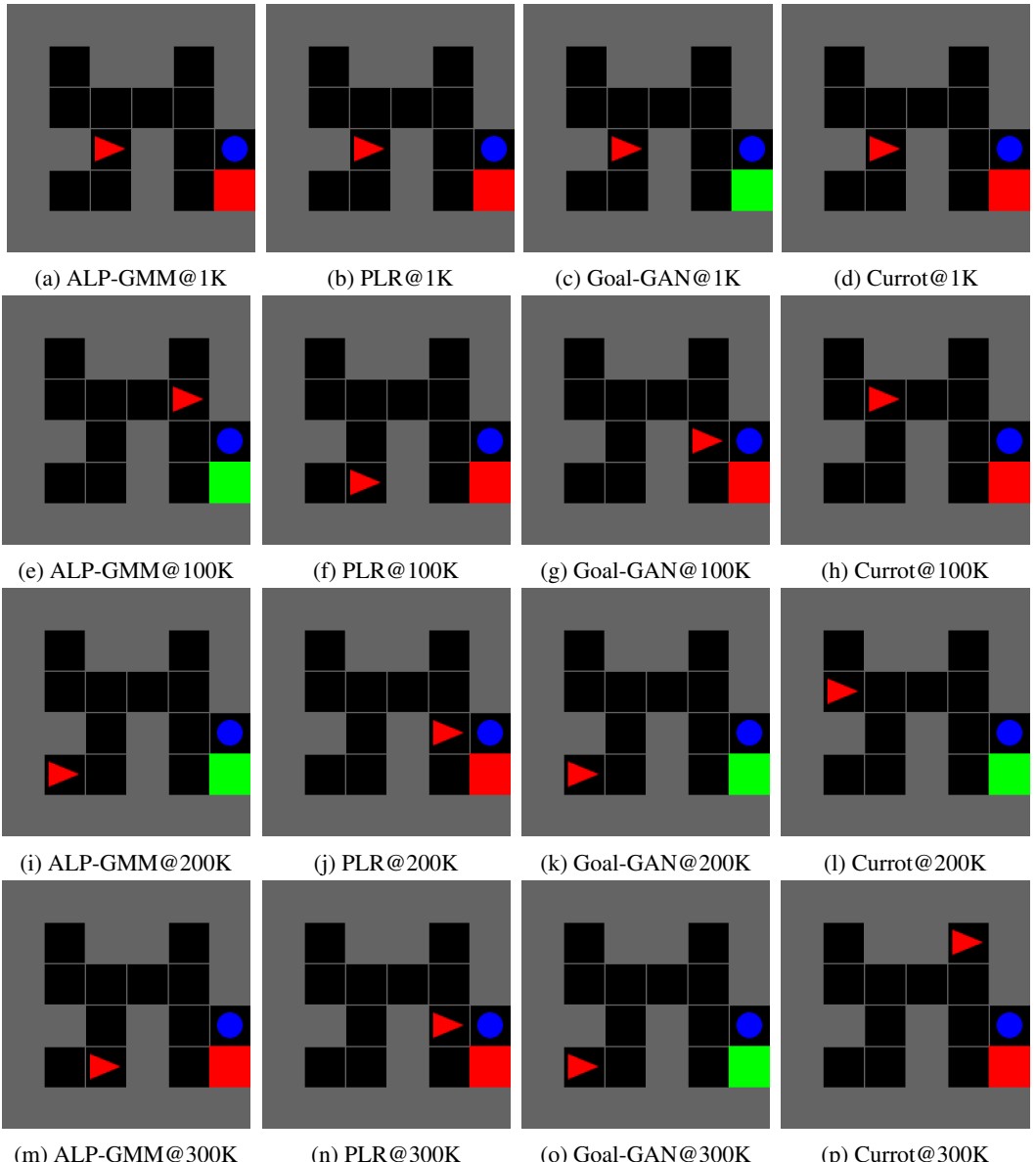

Figure 11: Curricula generated by the (causal) augmented curriculum generators for Button Maze. Each column shows a curriculum from one generator at different training steps.

literature (Correa & Bareinboim, 2020; Pearl & Bareinboim, 2011; Bareinboim & Pearl, 2016). The literature on transportability studies broadly how to answer queries with data from disparate domains. By examining shared causal mechanisms across seemingly dissimilar domains, formal graphical conditions are established to identify what queries can be answered and how those should be answered. In our curriculum reinforcement learning setting, a good curriculum elicits policy that performs well not only in subtasks but, more importantly, in target tasks. Policy pieces from various tasks in the curricula constitute our data from which our quest is to construct an optimal target task policy. Since subtasks are generated from target tasks, they share certain aspects in principle. Analyzing how we can transfer those policy pieces to the target tasks can thus be viewed as a transportability problem.

Throughout the paper, we assume access to the causal diagram of the task. However, there is also an orthogonal line of research dedicated in learning the causal structure directly from the task from which a causal diagram can be derived naturally (Hu et al., 2022; Li et al., 2023; Perry et al., 2022).

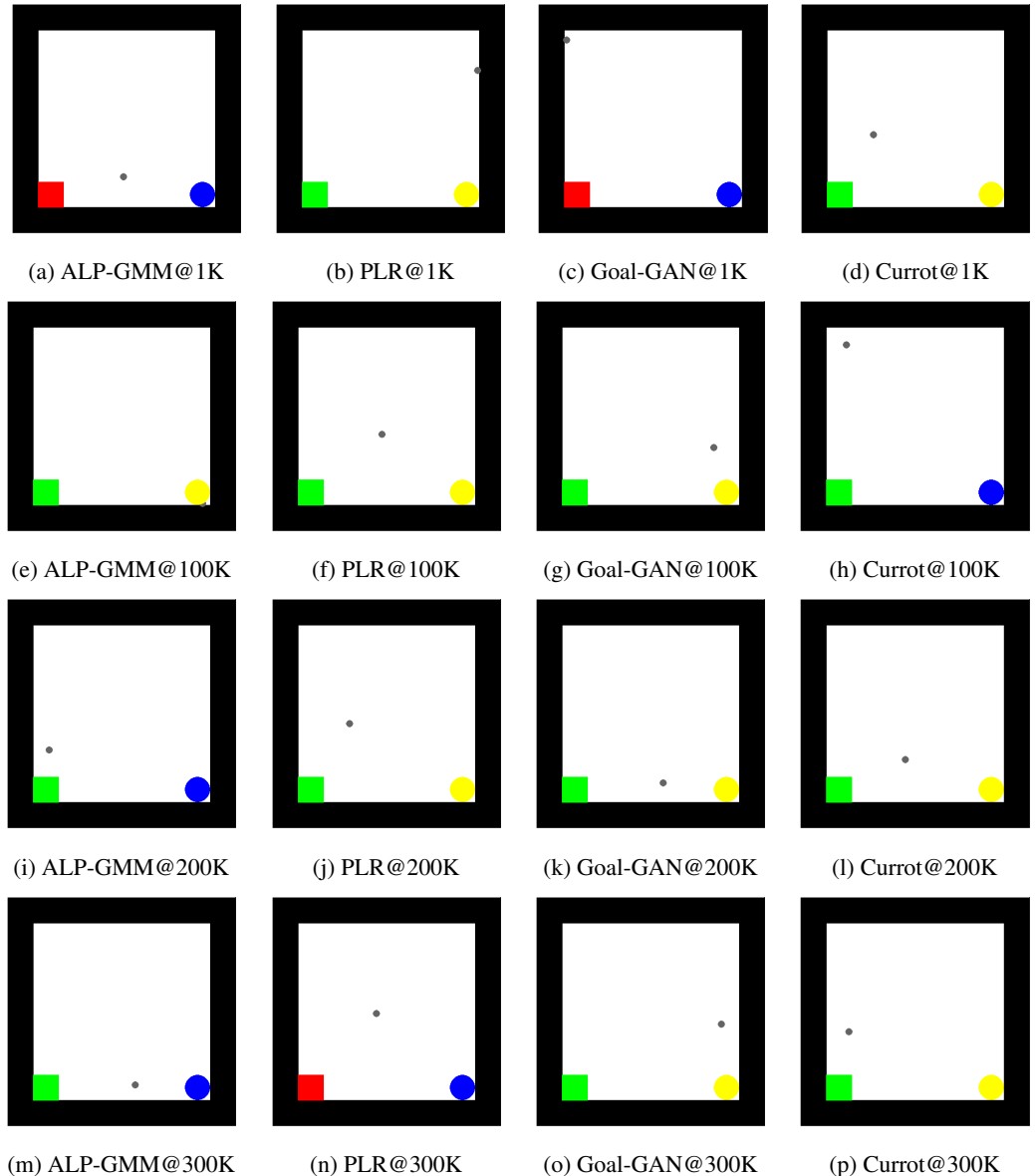

Figure 12: Curricula generated by the (causal) augmented curriculum generators for the Continuous Button Maze. Each column shows a curriculum from one generator at different training steps.

# H  MARKOV DECISION PROCESSES (MDPS), PARTIALLY OBSERVABLE MDPS (POMDPS), AND STRUCTURAL CAUSAL MODELS (SCMS)

An MDP is defined to be a four-tuple $\langle \mathcal{S}, \mathcal{A}, \mathcal{R}, T \rangle$ where $\mathcal{S}$ is a finite set of states, $\boldsymbol{A}$ a finite set of actions, $T : \mathcal{S} \times \mathcal{A} \to \Pi(S)$ the transition function mapping from state action pair to the distribution over the set of states and $\mathcal{R} : \mathcal{S} \times \mathcal{A} \to \mathbb{R}$ the reward function (Kaelbling et al., 1996). And a POMDP is defined to be a six tuple with two additional elements than the MDP, $\langle \mathcal{S}, \mathcal{A}, \mathcal{R}, T \rangle, \mathcal{O}, p$ where $\mathcal{O}$ is a finite set of observations and $p : \mathcal{S} \to \Pi(\mathcal{O})$ is the observation function mapping from the true underlying state to the distribution of observations (Kaelbling et al., 1998). Recall the definition of SCMs in the preliminary section, we can see that the definition of SCMs subsumes the transition function and inherent structural assumptions in MDPs and POMDPs. We can encode all the state/action/observation/reward variables as endogenous variables and the randomness of the

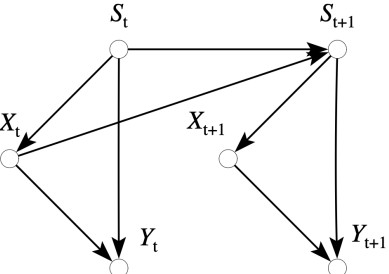

Figure 13: The causal diagram of a standard MDP. We use $X$ to denote actions and $Y$ to denote rewards.

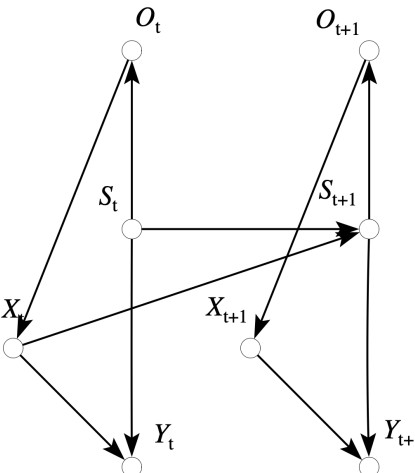

Figure 14: The causal diagram of a standard POMDP. We use $X$ to denote actions and $Y$ to denote rewards.

reward and transition functions as exogenous variables. Each variable's value is either determined by a structural equation specified by the environment or a policy specified by the agent.

Graphically speaking, the causal diagram of a typical MDP is shown in Fig. 13. The next state $S_{t+1}$ only takes the action and state at the current time step as input, which means that in the form of structural equations, $S_{t+1} = f_{S_{t+1}}(S_t, X_t, U_{S_{t+1}})$ where $U_{S_{t+1}}$ is an exogenous variable representing the inherent randomness in the transition function. In Fig. 13, we can see clearly the Markov assumption embedded inside the graphical structure, i.e., $(S_{t+1} \perp\!\!\!\perp S_{t-1}|S_t, X_t)$. Similarly, we can also ground the Markovian reward assumption with precise graphical criteria. As stated by Abel et al., Markovian reward assumption assumes that the state factors that are affecting the reward are fully observable to the agent. This can be interpreted as, $pa(Y_t) \subseteq S_t$ where all parent nodes of the reward node is given as input to the agent ($S_t$). Note that here we denote the state as a set of variables instead of one single variable for clarity and we implicitly assume that the agent can observe all $S_t$ when making decisions.

For POMDP, the causal diagram is shown in Fig. 14. The causal diagram faithfully reflects the fact that the agent cannot observe the state variables directly but only the observations. And importantly,

the underlying transition dynamics between the true states and actions are still following the standard MDP. The representation of SCMs is more versatile in modeling decision making scenarios in that it is amenable to represent any data generating processes without casting structural assumptions like MDP nor POMDP to the problem. By introducing the notion of confounders, we can better utilize the graphical structure to construct optimal agents efficiently (Zhang & Bareinboim, 2020).

In this work, we utilize the qualitative causal knowledge to ensure that the causal effect of changing certain aspects of the target task won't affect the optimal policy the agent will learn from the generated source task. Even without confounders, if the state space is partially observable, the same situation as in the Colored Sokoban could happen since not all factors affecting the reward can be observed by the agent. But when the Markovian reward assumption holds, where the agent can observe all the parents of the reward variable, the reward cannot be confounded with any other variables. Under this stronger assumption, as our Theorem 1 indicates, all state variables are editable. On the other hand, we are dedicated to solving the curriculum generation problem in the presence of unobserved confounders. Thus, the setting of Markovian rewards actually falls into the traditional curriculum reinforcement learning problem studied in the literature, which our work is trying to relax.

## I  LIMITATIONS

As our theorems and algorithms require causal diagrams as inputs, it might not be easily applicable when the domain specific knowledge is hard to retrieve or the input diagram is inaccurate. Thus, we will combine our method with causal discovery methods that can automatically learn the underlying causal structure from the data to loosen this constraint in the future (Hu et al., 2022; Li et al., 2023; Perry et al., 2022).

