# OpenReview forum: "Causally Aligned Curriculum Learning"
_ICLR.cc/2024/Conference — ICLR 2024 poster_

### Official Review · Reviewer_1poU · 2023-10-23

**Soundness:** 3 good
**Presentation:** 3 good
**Contribution:** 3 good
**Rating:** 8
**Confidence:** 3

**Summary:**

This paper presents a curriculum design method that produces causally aligned curriculums. Given a causal graph, a curriculum generator, and a target task, the proposed method creates a causally aligned curriculum using two main ideas. First, the FindMaxEdit algorithm, which constructs a set of causally aligned source tasks based on the modifying the editable variables of the target task. Importantly, optimal decision rules can be transported across causally aligned source tasks. Second, the authors propose ordering causally aligned source tasks with an expansion criterion on the optimal decision rules (i.e. tasks should be ordered such that the set of optimal decision rules expands with each task).

**Strengths:**

i. The method is well-founded and rigorously specified. Further, the main idea of this paper - using knowledge of causal relationships to improve curriculum generation methods - is clearly useful to the community.
ii. The paper is clear, with all details necessary to reproduce the work present in the paper.
iii. The experimental results on Colored Sokoban and Button Maze are strong.

**Weaknesses:**

Most of my concerns + comments are related to the motivation + presentation of the paper.

- Motivation: Example 1's motivation for the paper is very contrived. The only reason a curriculum designer would design such a source task (where the box color is fixed to yellow) is if the curriculum designer was completely unaware of the causal dependencies of the reward. While this is possible in the case of causally-unaware automated curriculum generation methods, I find it a bit unlikely unless the crucial confounder variable U_i was either excluded from the state space (which would violate the usual Markovian reward assumption) or it is included in teh state space, but the agent cannot observe it (a partial observability assumption).  This leads to my questions:
	1)  Can the authors experimentally verify how often existing non-causal curriculum generation methods generate causally unaligned tasks?
	2) Can the authors comment on the expected benefit of their method in a domain with Markovian reward, or a fully observable domain w/no unobserved confounders?

-  Acknowledging limitations and situating their work in presence of related work
	1) I saw that the related works section was relegated to the Appendix. There should at least be a pointer in the main paper to the related works. I found it helpful to read it, to understanding the positioning of this work, and think that it should be included in the main paper. The paper provides almost too many details about the method in the main paper, and at places is rigorous to the point of being pedantic. I think that the space would be better spent on situating the work properly.
	2) The inputs/outputs of the authors' method is not presented in plain language. As I understand it, the authors' method assumes access to the SCM of the task, and a curriculum generator such as GoalGAN. Such details should be made much clearer, perhaps through a summary methods figure or by adding a discussion of limitations.

- Figures are not standalone: it is common practice to read a paper by skimming the figures + captions first, but I found that it was not possible to get the main idea of the figure, method and experimental results by doing so. Even after reading most of the accompanying text, I still found the figures + captions alone ambiguous. I had to find the specific part of the text referencing the figures -- sometimes needing to jump to other parts of the paper-- and read very carefully to understand what was happening. The authors should rewrite the figure captions and perhaps add a new "methods" figure summarizing the flow of their method. My specific comments on two figures are below, but all figures can be improved:
	1) Figure 5: Specifically, I was confused about why the curves labelled "causal" performed differently across the four columns? Also, the names of "causal" vs "original" were confusing. I think this is because the fact that the proposed method augments existing curriculum generation methods was presented only in a single sentence towards the end of the intro, and the knowledge was assumed in the rest of the paper. Seeing as this information is crucial to understand the paper, perhaps the authors can add this crucial piece
	2) Figure 3: It's confusing that figure (a) and (b) are identical except for the edit indicators. I needed to visually trace all arrows of both figures to verify this. The presence/absence of edit indicators should be explained in the caption, and the figure can be modified so that it is much quicker for the reader to notice that the only addition is the edit indicators (perhaps through the use of opacity / shading).

**Questions:**

See the weaknesses section for the most substantial questions and comments.

Minor clarity comments are below:
- Exogenous vs endogenous should be defined at some point in the preliminaries section on SCMs.
- "che" "de" "an" meanings were not immediately clear to me.
- In Def 1, the SCM M* is labelled with a *, yet this notation does not appear elsewhere in the paper.

---

> ### Author Response · Authors · 2023-11-21
>
> Dear reviewer 1poU, thank you for your detailed suggestions and constructive feedback. Our work resides in a multidisciplinary field, and it is challenging to motivate the work in a way that is amenable to both sides, reinforcement learning and causal inference researchers. Your insighful comments will greatly assist in this endeavor. We will try incorporate your feedback and the subsequent discussion into our paper.
>
> >#### Weakness 1:
> >“Motivation: Example 1's motivation for the paper is very contrived. The only reason a curriculum designer would design such a source task (where the box color is fixed to yellow) is if the curriculum designer was completely unaware of the causal dependencies of the reward. While this is possible in the case of causally-unaware automated curriculum generation methods, I find it a bit unlikely unless the crucial confounder variable U_i was either excluded from the state space (which would violate the usual Markovian reward assumption) or it is included in the state space, but the agent cannot observe it (a partial observability assumption). This leads to my questions:
> >1.  Can the authors experimentally verify how often existing non-causal curriculum generation methods generate causally unaligned tasks?
> >2.  Can the authors comment on the expected benefit of their method in a domain with Markovian reward or a fully observable domain w/o unobserved confounders?”
>
> Recall that in the definition of SCMs, each variable $V$’s value is determined by a function of $f_V(\operatorname{pa}(V), \boldsymbol{U}_V)$ where $\operatorname{pa}(V)$ are the values of parents of $V$ and $\boldsymbol{U}_V$ are exogenous variables. The inherent randomness of the system is due to the existence of exogenous variables. When an exogenous variable affects more than one endogenous variable, it is called a confounder\, which is unobservable to the agent and the curriculum designer. This holds for both automatic curriculum generation and manual curriculum designing. While it is a common practice to assume Markovian rewards or fully observable state space, as we stated in the introduction, our work investigates a more challenging scenario where the underlying environment contains unobserved confounders. For instance, in example 1 (the Colored Sokoban), one unobserved confounder is $U_i$, which is affecting both the box color $C_i$ and the reward $Y_i$. The randomness of both $C_i, Y_i$ are determined by $U_i$.
>
> 1.1 Please see the following chart for the percentage of misaligned source tasks generated by the baseline methods we tested in the both environments during training (300K environment steps). We can see that the existing non-causal curriculum generators are indeed generating a large portion of misaligned source tasks during the training even though they have the option to choose aligned source tasks. This indicates that the existing non-causal curriculum generators cannot effectively avoid misaligned source tasks and thus they are detrimental to the agent training when applied to environments with unobserved confounders.
>
>
> |Environments/Curriculum Generators |ALP-GMM|PLR|Currot|Goal-GAN
> |--|--|--|--|--|
> |**Colored Sokoban**|68.10%$\pm$0.71%|69.10%$\pm$2.77%|68.36%$\pm$1.06%|66.41%$\pm$1.84%|
> |**Button Maze**    |88.41%$\pm$0.43%|87.33%$\pm$3.77%|88.62%$\pm$1.00%|90.30%$\pm$0.27%|
>
> 1.2 Even without confounders, if the state space is partially observable, the same situation as in the Colored Sokoban could happen. When the Markovian reward assumption holds, where the agent can observe all the parents of the reward variable, the reward cannot be confounded with any other variables. Under this stronger assumption, as our Theorem 1 indicates, all state variables are editable. On the other hand, we are dedicated to solving the curriculum generation problem in the presence of unobserved confounders. Thus, the setting of Markovian rewards actually falls into the traditional curriculum reinforcement learning problem studied in the literature, which our work is trying to relax.

---

> > ### Author Response · Authors · 2023-11-21
> >
> > >#### Weakness 2:
> > >“Acknowledging limitations and situating their work in presence of related work
> > >1.  I saw that the related works section was relegated to the Appendix. There should at least be a pointer in the main paper to the related works. I found it helpful to read it, to understanding the positioning of this work, and think that it should be included in the main paper. The paper provides almost too many details about the method in the main paper, and at places is rigorous to the point of being pedantic. I think that the space would be better spent on situating the work properly.
> > >2.  The inputs/outputs of the authors' method is not presented in plain language. As I understand it, the authors' method assumes access to the SCM of the task, and a curriculum generator such as GoalGAN. Such details should be made much clearer, perhaps through a summary methods figure or by adding a discussion of limitations.”
> >
> > 2.1 Thank you for the suggestion; we have added a pointer in the introduction section to the related work. We have discussed how our work is related to the literature in the introduction section, a condensed version of the related work section in the appendix. Given the theoretical nature of the work, we strive for a precise and self-contained presentation of the results. Due to space limitations, we only kept the most closely related literature for positing and motivating our work in the introduction section. We have provided an extended version of the related work in the appendix G.
> >
> > 2.2 The plain text descriptions of input/output of algorithms 1 and 3 can be found in the paragraph beside the algorithm pseudo-code, which is also shown in each pseudo-code box's input/output field. We would also like to clarify that SCMs are *not* assumed as inputs. Instead, we only require causal diagrams as inputs, a graphical representation of the dependency relationships among the variables without any detailed causal mechanisms or transition probabilities (the F and P(u) in the definition of SCM in the preliminary section).
> >
> > >#### Weakness 3:
> > >“Figures are not standalone: it is common practice to read a paper by skimming the figures + captions first, but I found that it was not possible to get the main idea of the figure, method and experimental results by doing so. Even after reading most of the accompanying text, I still found the figures + captions alone ambiguous. I had to find the specific part of the text referencing the figures -- sometimes needing to jump to other parts of the paper-- and read very carefully to understand what was happening. The authors should rewrite the figure captions and perhaps add a new "methods" figure summarizing the flow of their method. My specific comments on two figures are below, but all figures can be improved:
> > >1.  Figure 5: Specifically, I was confused about why the curves labelled "causal" performed differently across the four columns? Also, the names of "causal" vs "original" were confusing. I think this is because the fact that the proposed method augments existing curriculum generation methods was presented only in a single sentence towards the end of the intro, and the knowledge was assumed in the rest of the paper. Seeing as this information is crucial to understand the paper, perhaps the authors can add this crucial piece
> > >2.  Figure 3: It's confusing that figure (a) and (b) are identical except for the edit indicators. I needed to visually trace all arrows of both figures to verify this. The presence/absence of edit indicators should be explained in the caption, and the figure can be modified so that it is much quicker for the reader to notice that the only addition is the edit indicators (perhaps through the use of opacity / shading).”
> >
> > Thanks for your suggestion! We agree that the caption should be self-contained for ease of skimming. We have improved them and added a method flow chart to the main text.
> >
> > 3.1 We agree that the caption in Figure 5 doesn’t stand alone, and we have improved the wording in the revised version. Still, we would like to clarify that we also have provided detailed descriptions of the figure in the experiment section that we compare the performance between “the original curriculum generators” and “the causal version of these generators” using “four best-performing curriculum generators” (four columns).
> > In Figure 5, “causal” refers to the augmented version of the baseline curriculum generators, and “original” refers to the unaugmented, original version of the baseline curriculum generators. The first row is the performance in the Colored Sokoban, and the second row is the performance in the Button Maze. Each column is a different curriculum generator we tested.
> >
> > 3.2 Thanks for pointing this out, we have adjusted the figure.

---

> > > ### Author Response · Authors · 2023-11-21
> > >
> > > >#### Question 1:
> > > >“Exogenous vs endogenous should be defined at some point in the preliminaries section on SCMs. ”
> > >
> > > Please see line 3 of page 3 for the definitions of exogenous vs endogenous; happy to provide further clarifications if needed.
> > >
> > > >#### Question 2:
> > > >“"che" "de" "an" meanings were not immediately clear to me. ”
> > >
> > > We mentioned in the last paragraph of the preliminary section that we use standard graph-theoretic family abbreviations to represent graphical relationships, such as parents, children, descendants, and ancestors. Thus, in this context, “$Ch(X)$” refers to children of a variable $X$, which are the nodes that $X$ has a direct edge into. “$De(X)$” refers to the descendants of an variable $X$, which are all the nodes that $X$ has a directed path into. “$An(X)$” refers to all the ancestors of $X$ which have a directed path into $X$. We have also improved the wording to reflect this more clearly in the revised version.
> > >
> > > >#### Question 3:
> > > >“In Def 1, the SCM M* is labelled with a *, yet this notation does not appear elsewhere in the paper.”
> > >
> > > Thanks for noting this. We have removed the * notation in the updated version.

---

> > > > ### Comment · Reviewer_1poU · 2023-11-21
> > > >
> > > > Thanks for the authors' replies! I appreciated them and found that the modifications improved my understandinig of the paper. Since my opinion that the paper should be accepted remains unchanged, I will not change my score.

---

> > > > > ### Author Response · Authors · 2023-11-22
> > > > >
> > > > > Thank you! We are glad that our response helps in clarifying your concerns!

---

### Official Review · Reviewer_eG2G · 2023-10-30

**Soundness:** 3 good
**Presentation:** 4 excellent
**Contribution:** 2 fair
**Rating:** 6
**Confidence:** 3

**Summary:**

The paper tackled the problem of curriculum design in multi-task Reinforcement Learning. A challenge in the previous curriculum learning literature is that they all assume that the optimal decision rules are shared across different tasks. From a causal point of view, this expectation may not hold when the underlying environment contains unobserved confounders. To tackle this issue, the paper proposed a causal framework based on structural causal models and rigorously defined the notion of aligned tasks. The paper then proposed an algorithm that only generates aligned tasks. Simulation studies on Maze and Sokoban environments showed the advantage of the proposed algorithms.

**Strengths:**

The paper finds a significant issue in the current literature of automatic curriculum learning and provides a clear framework to discuss the issue of non-alignment. Thus, the paper has great significance.

The paper is well-written and all the important messages are clearly delivered.

**Weaknesses:**

1. The authors mentioned that curriculum learning could overcome the curse of dimensionality. However, all the examples discussed in this paper are tabular. There seems to be a mismatch. It would be interesting to discuss and experiment on an example with exponentially large state and action space.

2. What is the computation-complexity of FindCausalCurriculum? Does it scale with the size of state and action space? If that is the case, it seems to be against the motivation of applying curriculum learning.

3. Comments on experiments: In general, I believe the environment tested in this paper are rather naive.

The authors implement ALP-GMM by fixing the color. I don't think this is a good way to construct baseline. One can run ALP-GMM as a task sampler on a fixed set of tasks. ALP-GMM essentially assigns a probability for different tasks at the each round. If one task has the underlying U that is not aligned with the target task, ALP-GMM will adaptively reduce the weight for that task.

I understand that fixing color matches the general story of the whole paper. However, I don't think this is a natural choice in examples where we have access to a simulator. It would be interesting to demonstrate a non-trivial example where C is a proxy of the underlying U and choosing C by running ALP-GMM could be harmful.

4. An important way to improve the paper is to propose an algorithm for large (or continuous) state and action space. The current version seems to only work with tabular MDPs, which limits the complexity of environments the algorithm can be applied to.

**Questions:**

1. I was a bit confused by the wordings in abstract. The authors first state that invariant
optimal decision rules does not necessarily hold, then they propose condition characterizing causally aligned source tasks, i.e., the invariance of optimal decision rules holds. The flow does not seem right here.

2. Need extra clarifications on Figure 1 and 2. What does it mean by fixing the color? When the different algorithms are trained, are they only allowed to choose tasks from misaligned source tasks? Should this example be too artificial? It would be interesting to see the results when ALP-GMM can generate tasks with unknown color, while the algorithm may reuse these tasks. In this way, it is possible for ALP-GMM to discover that certain tasks can be more helpful for learning the target task. This is in fact more aligned with the situations in the real-world.

---

> ### Author Response · Authors · 2023-11-21
>
> Dear reviewer eG2G, thank you for your detailed review. We appreciate your acknowledgment of the significance of our work and have addressed each of your concerns in the following responses.
>
> >#### Weakness 1:
> >“The authors mentioned that curriculum learning could overcome the curse of dimensionality. However, all the examples discussed in this paper are tabular. There seems to be a mismatch. It would be interesting to discuss and experiment on an example with exponentially large state and action space.”
>
> We agree that the examples are in the grid world, but they are only for illustration purposes and don’t indicate any limitations of our proposed method. For the ease of understanding, we delibrately choose such simple domains to better demonstrate our ideas and the implications of unobserved counfounders in the environment. One of the initial observations in these settings is that applying standard non-causal curriculum generators could be detrimental. Given that the environment is simple, we were able to explain the subtle interplay between curriculum generation, decision-making and unobserved confounding.
>
> Having said, we note that our technical results (e.g., theorems, algorithms) are non-parametric, which means that they are immediately applicable to high-dimensional, continuous domains given merely qualitative causal knowledge about the domain. Once the proposed augmentation procedure is executed, we can leverage the computational power of existing curriculum generation methods to address those more complex domains. We also note that the observation spaces in our experiments are all in raw pixels instead of low-dimensional state vectors, which are high-dimensional. From the performance of agents trained directly in the target task (figure 5, horizontal green line), we can see that without causal curriculum learning, those agents cannot converge to the optimal in the target task at all, which suggests that our method helps with the learning.
>
> Finally, and based on the feedback provided, we further conducted an extra experiment with continuous actions and states to demonstrate the effectiveness of our methods in exponentially large state and action space empirically. Please see the common response to all the reviewers for the detailed results.
>
> >#### Weakness 2:
> >“What is the computation-complexity of FindCausalCurriculum? Does it scale with the size of state and action space? If that is the case, it seems to be against the motivation of applying curriculum learning.”
>
> The worst-case time complexity of FindCausalCurriculum is $\mathcal{O}(Hdn^2)$ where $H$ is the length of the horizon, $d$ is the number of actions, $n$ is the total number of nodes in the causal diagram. In an improved version of FindCausalCurriculum, which we show in the appendix (Algo 6 Causal Curriculum Learning), the worst-case time complexity is reduced to $\mathcal{O}(Hn^2)$. Since there is no free lunch, given the performance boost and this relatively slow scaling overhead w.r.t the state/action size, applying causal curriculum learning is still preferable. More specifically, as shown in Figure 5, without curriculum learning, the agent trained directly in the target task cannot converge to the optimal at all. Even with curriculum learning, failing to utilize causal knowledge results in worse performance.

---

> > ### Author Response · Authors · 2023-11-21
> >
> > >#### Weakness 3 & 4:
> > >"Comments on experiments: In general, I believe the environment tested in this paper are rather naive.”
> > >”The authors implement ALP-GMM by fixing the color. I don't think this is a good way to construct baseline. One can run ALP-GMM as a task sampler on a fixed set of tasks. ALP-GMM essentially assigns a probability for different tasks at the each round. If one task has the underlying U that is not aligned with the target task, ALP-GMM will adaptively reduce the weight for that task.
> > I understand that fixing color matches the general story of the whole paper. However, I don't think this is a natural choice in examples where we have access to a simulator. It would be interesting to demonstrate a non-trivial example where C is a proxy of the underlying U and choosing C by running ALP-GMM could be harmful.”
> > >“An important way to improve the paper is to propose an algorithm for large (or continuous) state and action space. The current version seems to only work with tabular MDPs, which limits the complexity of environments the algorithm can be applied to.”
> >
> > We agree that these two environments may seem relatively naive from a human perspective. However, this fact actually strengthens the message of our paper. Namely, if deep reinforcement learning agents cannot solve such seemingly simple problems perfectly (figure 5, trained directly in target task) and if using curriculum learning without incorporating causal knowledge can sometimes be detrimental, how can we expect these agents to achieve similar  goals in more challenging environments?
> >
> > For the ALP-GMM experiments, we would like to clarify that we didn’t force any of the curriculum generators to fix the box color. In the context space, all curriculum generators have the option to “not fix the box color.” Thus, they are already running under your suggested scenario where “C is a proxy of the underlying U, and choosing C by running ALP-GMM (also true for other baseline curriculum generators) could be harmful.” You can also see the following chart summarizing the portion of misaligned source tasks against aligned source tasks during training.
> >
> > |Environments/Curriculum Generators |ALP-GMM|PLR|Currot|Goal-GAN
> > |--|--|--|--|--|
> > |**Colored Sokoban**|68.10%$\pm$0.71%|69.10%$\pm$2.77%|68.36%$\pm$1.06%|66.41%$\pm$1.84%|
> > |**Button Maze**    |88.41%$\pm$0.43%|87.33%$\pm$3.77%|88.62%$\pm$1.00%|90.30%$\pm$0.27%|
> >
> > The reason those curriculum generators fail to avoid such harmful source tasks adaptively is that there is still a non-trivial chance (in this setting, $P(U=1)=1/4$) that the agent can get rewarded for pushing an intervened blue color box (undesired behavior). There is also a high probability ($P(U=0)=3/4$) that the agent gets punished for pushing an intervened yellow color box (desired behavior).
> >
> > Our theorems/algorithms are non-parametric, which means we don’t restrict the parametric form or the size of the scope of those state/action variables neither the distribution over exogenous conditions. They readily apply to high-dimensional (large) or continuous domains. From the experiments, all observations are in the pixel space, which is high-dimensional. Also, note that the environments we tested are not MDP due to the existence of confounders.
> >
> > Our method is an augmentation to the existing curriculum generators, preparing them for domains where unobserved confounding cannot be ruled out a priori. It does not contradict but rather complements these generators, and is applicable in the same environments as they can support. We agree that showing an experiment in a continuous domain would make the contributions of the paper clearer. Therefore, we have included an extra experiment featuring continuous state and action variables to empirically demonstrate the effectiveness of our method. For the results, please refer to the common response to all the reviewers.

---

> > > ### Author Response · Authors · 2023-11-21
> > >
> > > >#### Question 1:
> > > >“I was a bit confused by the wordings in abstract. The authors first state that invariant optimal decision rules does not necessarily hold, then they propose condition characterizing causally aligned source tasks, i.e., the invariance of optimal decision rules holds. The flow does not seem right here.”
> > >
> > >
> > > We appreciate the opportunity to clarify this issue. In the abstract, we say that the expectation of applying curriculum learning is for the optimal decision rules to remain invariant across source tasks and the target task. However, this expectation does not always hold, particularly when the underlying state variables are factorized and unobserved confounders exist. To address this, we study this challenge in depth and propose formal conditions of when this expectation is met or not. This analysis enables us to extend existing curriculum learning algorithms to a broader range of environments where unobserved confounding is prevalent (Recall that by the definition of SCMs, endogenous variables are confounded if they share the same exogenous variables as inputs in determining their values).
> > >
> > >
> > >
> > > >#### Question 2:
> > > >“Need extra clarifications on Figure 1 and 2. What does it mean by fixing the color? When the different algorithms are trained, are they only allowed to choose tasks from misaligned source tasks? Should this example be too artificial? It would be interesting to see the results when ALP-GMM can generate tasks with unknown color, while the algorithm may reuse these tasks. In this way, it is possible for ALP-GMM to discover that certain tasks can be more helpful for learning the target task. This is in fact more aligned with the situations in the real-world.”
> > >
> > >
> > > “Fixing the color” refers to the misaligned source tasks where the box color is intervened. The example in Figure 1 and 2 is simplified for illustration purposes, it does not indicate the limitations of our approach. However, this example is not too artificial at all because during the experiments, all curriculum generators are free to choose any context, including fixing the box color (misaligned) and not fixing the box color (aligned). This means that learners are trained in a combination of both aligned and misaligned source tasks. In the above chart (in response to Weakness 3&4), we counted the portion of misaligned source tasks generated by an unaugmented ALP-GMM and other curriculum generators we tested.
> > >
> > > We can see that they all fail to avoid those misaligned source tasks. As a result, agents trained by them fail to achieve the same convergence rate as the optimal agent trained only in the aligned curriculum. We have updated the captions in the revised version; thank you.
> > >
> > > Overall, the challenge of misaligned source tasks cannot be addressed by changing the distribution of the box color. This means even for a source task induced by a stochastic intervention on the box color, and its distribution coincides with the box color distribution in the target task, the misalignment could still occur. For concreteness, consider Example 1, instead of using do-interventions, we apply a stochastic intervention $\sigma_{C_i}$  to the box color such that $P(C_i = 0|\sigma_{C_i}) = \frac{1}{4}$ and $P(C_i = 1|\sigma_{C_i}) = \frac{3}{4}$. Under this intervention, the observational distribution of box color is exactly the same as the one from the target task. However, the conditional reward distributions that the policy is optimized towards are drastically different. In the target task, we have,
> > > $$P\left(Y_i = 10 \big| B_i =  \text{next to goal}, C_i, \operatorname{do}\left(X_i = \text{push}\right)\right) = 1[C_i = 0]$$, and
> > > $$P\left(Y_i = -10 \big| B_i =  \text{next to goal}, C_i, \operatorname{do}\left(X_i = \text{push}\right)\right) = 1[C_i=1]$$.
> > > But in the intervened world, by sigma-calculus [1], we have,
> > > $$P\left(Y_i = 10 \big| B_i =  \text{next to goal}, C_i, \operatorname{do}\left(X_i = \text{push}\right); \sigma_{C_i}\right) = \frac{1}{4}$$, and
> > > $$P\left(Y_i = -10 \big| B_i =  \text{next to goal}, C_i, \operatorname{do}\left(X_i = \text{push}\right); \sigma_{C_i}\right) = \frac{3}{4}$$.
> > > Thus, the optimal policy we learned in this source task still intends to not push the box at all while the optimal target task policy should be to push the yellow box.
> > >
> > > [1] Correa, Juan, and Elias Bareinboim. "A calculus for stochastic interventions: Causal effect identification and surrogate experiments." Proceedings of the AAAI conference on artificial intelligence. Vol. 34. No. 06. 2020.

---

### Official Review · Reviewer_StU4 · 2023-10-31

**Soundness:** 2 fair
**Presentation:** 3 good
**Contribution:** 2 fair
**Rating:** 3
**Confidence:** 4

**Summary:**

Curriculum reinforcement learning is to train the agent on a sequence of simpler, related tasks in order to gradually progress the learning towards a difficult, often sparse-reward target task. The hope is that if there are common optimal decisions among these simpler intermediate tasks and the target task, the agent will learn transferrable skills during the process and accelerate the learning process. However, this assumption may not hold if there are unobserved confounders in the environment. This paper delves into this problem from a causal perspective, defining conditions under which the optimal decisions remain consistent. It also introduces a method to create a curriculum that aligns causally. The method is tested in two grid-world environments with unseen confounders to validate the effectiveness of the proposed algorithms.

**Strengths:**

The structure and flow of the paper are easy to follow. Several simplified examples are provided to support the argument and illustration. To my knowledge, the perspective of confounders in curriculum reinforcement learning is novel.

**Weaknesses:**

* Toyish experiments. The experiments conducted are limited to grid-world environments, which significantly narrows the scope of application. For the conclusions to be generalized, the experiments should ideally be extended to a more diverse set of environments.
* Scalability. The paper does not adequately address how the proposed method scales to continuous environment variables. This is a significant concern as many practical applications in continuous or high-dimensional environment variable space. By not tackling this challenge, the authors leave a gap in understanding the full potential and limitations of their method.
* Missing related work:
    * Hu, Xing, et al. "Causality-driven Hierarchical Structure Discovery for Reinforcement Learning." Advances in Neural Information Processing Systems 35 (2022): 20064-20076.
    * Cho, Daesol, Seungjae Lee, and H. Jin Kim. "Outcome-directed Reinforcement Learning by Uncertainty & Temporal Distance-Aware Curriculum Goal Generation." arXiv preprint arXiv:2301.11741 (2023).
    * Huang, Peide, et al. "Curriculum reinforcement learning using optimal transport via gradual domain adaptation." Advances in Neural Information Processing Systems 35 (2022): 10656-10670.

**Questions:**

* In Algorithm 3, how is the actions sequence $\{X_1, \ldots, X_H\}$ obtained in the first place? If we have this optimal action sequence before the learning starts, why do we need RL?
* In Theorem 3, one of the conditions is that: For every $j=1, \ldots, N-1$, actions $\boldsymbol{X}^{(j)} \subseteq \boldsymbol{X}^{(j+1)}$. Does this mean that the transition must be deterministic and there only exists one unique optimal solution?

---

> ### Author Response · Authors · 2023-11-21
>
> Dear reviewer StU4, thank you for your comments and feedback. We believe that a few misreadings of our work made some of the evaluations overly harsh and would like to ask you to reconsider the proposed paper in light of the clarifications provided below.
>
> > ####  Weakness 1&2:
> > “Toyish experiments. The experiments conducted are limited to grid-world environments, which significantly narrows the scope of application. For the conclusions to be generalized, the experiments should ideally be extended to a more diverse set of environments.
> > “Scalability. The paper does not adequately address how the proposed method scales to continuous environment variables. This is a significant concern as many practical applications in continuous or high-dimensional environment variable space. By not tackling this challenge, the authors leave a gap in understanding the full potential and limitations of their method.”
>
> This paper provides an augmentation procedure that allows existing curriculum generation algorithms to handle  environments with unobserved confounders. This augmentation procedure (Theorems 1-3) is non-parametric, relying on qualitative knowledge about the underlying environment. Once this augmentation procedure is executed, we can leverage the computational power of existing curriculum generation methods to address more complex domains. Consequently, our proposed method is generalizable to other, more diverse, high-dimensional/continuous environments.
>
> After all, we are aware that scaling to high-dimensional domains is an important challenge. However, this paper investigates an orthogonal challenge that arises due to the existence of unobserved confounding bias, which we believe is also important in the design of curriculum generators. The proposed approach does not contradict but complements existing curriculum learning methods, by generalizing them to a more diverse set of environments where confounding bias may be present.  (Example 1 illustrates how current curriculum generators could fail when applied to environments with unobserved confounding, which motivated this work.)
>
> Also, we would also like to note that our experiments use raw pixels as the observational space, which is high-dimensional, as stated in the experimental section and the conclusions. Still, following the reviewers’ concerns, we included an additional experiment for continuous state/action variables to illustrate the effectiveness of the proposed method empirically. Further details can be found in the common response to all the reviewers.
>
> >#### Weakness 3:
> >“Missing related work:
> >-   Hu, Xing, et al. "Causality-driven Hierarchical Structure Discovery for Reinforcement Learning." Advances in Neural Information Processing Systems 35 (2022): 20064-20076.
> >-   Cho, Daesol, Seungjae Lee, and H. Jin Kim. "Outcome-directed Reinforcement Learning by Uncertainty & Temporal Distance-Aware Curriculum Goal Generation." arXiv preprint arXiv:2301.11741 (2023).
> >-   Huang, Peide, et al. "Curriculum reinforcement learning using optimal transport via gradual domain adaptation." Advances in Neural Information Processing Systems 35 (2022): 10656-10670.”
>
> Thank you for the suggested references; we have now incorporated them into the related work section. After reading them, we would like to contextualize their relevance to our work. In particular, Paper 1 focuses on causal discovery learning the underlying data generating mechanism directly. In the future, we may explore incorporating such methods to learn causal diagrams from data, rather than assuming them as an input. Papers 2 and 3, which are both curriculum generators like those we have tested, align with our theorem. They can be augmented to be robust to, and function in the presence of unobserved confounders. Overall, these references provide interesting perspectives that  complement our work, which develops the foundations for curriculum generation in more generalized settings where unobserved confounders are present.

---

> > ### Author Response · Authors · 2023-11-21
> >
> > >#### Question 1:
> > >“In Algorithm 3, how is the actions sequence $X_1, \dots, X_H$ obtained in the first place? If we have this optimal action sequence before the learning starts, why do we need RL?”
> >
> > Action sequence $X_1, \dots, X_H$ follows the topological ordering of these variables in the environment. This representation is common in RL literature. For instance, consider a finite-horizon POMDP with H steps; its actions are ordered as $X_1, X_2, \dots, X_H$, following the time steps that the system dynamics unroll. In other words, in POMDP models, the topological ordering of actions is fixed a priori. RL algorithms are used to find an optimal policy determining the values of these actions.
> >
> >
> >
> > >#### Question 2:
> > >“In Theorem 3, one of the conditions is that: For every $j=1,\dots,N-1$, actions $\boldsymbol{X}^{(j1)}\subseteq \boldsymbol{X}^{(j+1)}$. Does this mean that the transition must be deterministic and there only exists one unique optimal solution?”
> >
> > The set of actions $\boldsymbol{X}^{(j)}$ is a set of random variables. Each action in this set is still a random variable, not a specific value. More specifically, condition (i) in Theorem 3 requires that each source task must be causally aligned with a set of actions that is the set of actions whose optimal decision rules are the same as those in the target task. Then, to avoid forgetting (Fig. 4), condition (ii) further requires that the curriculum orders those source tasks in a way that the set of causally aligned actions is expanding along following the curriculum. Combined together, we have the conditions for causally aligned curricula that is implemented as an algorithm in Algo. 3.
> >
> > In general, Theorem 3 identifies an effective ordering of source tasks so that the learner can converge to the optimal target policy without forgetting useful skills (Fig. 4).  However, the transition distribution in each source task is not deterministic but stochastic, and it does not assume a unique optimal solution. Thus, this condition doesn’t indicate that the transition is deterministic or there only exists one unique optimal solution. We hope this clarifies the issue.

---

### Official Review · Reviewer_C4uc · 2023-11-04

**Soundness:** 3 good
**Presentation:** 3 good
**Contribution:** 3 good
**Rating:** 6
**Confidence:** 3

**Summary:**

The paper delves into the challenges of designing a curriculum of source tasks to tackle a complex target task in the presence of unobservable confounding variables within the environment. The authors leverage the structural causal model framework (Pearl, 2009) to define causally aligned source tasks for a given target task. They propose a causally aligned curriculum that incorporates qualitative causal knowledge of the target environment, and they validate their approach through experiments in two confounded environments.

**Strengths:**

The paper addresses an important problem in curriculum reinforcement learning, highlighting the potential negative impact of inadequate task space design on target task performance.

The writing quality is commendable, even though the paper features heavy notation due to its nature. The repeated use of the Sokoban game aids comprehension and clarity regarding the paper's contributions and claims.

**Weaknesses:**

The strategy for avoiding misaligned source tasks, as proposed in this work, relies on causal knowledge about the underlying data-generating mechanisms within the environment. This requirement limits the broader applicability of the strategy, making it dependent on domain-specific knowledge. The practicality of obtaining a causal diagram G for a general target task remains uncertain.

It remains unclear whether the proposed causal curriculum strategy can be extended to domains with continuous state and action spaces.

**Questions:**

Re. the Colored Sokoban example:

In the baseline, the context/task space is restricted to tasks where the box color remains constant throughout the game. However, considering a more extensive task space that encompasses all possible combinations of box colors might lead the state-of-the-art curriculum strategies to automatically select relevant/aligned tasks during training. In contrast, the proposed causal curriculum limits the task space to initial agent and box positions, with the environment determining box colors based on intrinsic randomness.

Formally, let us denote the initial positions of the agent and the box as $(a_0^x, a_0^y)$ and $(b_0^x, b_0^y)$, respectively. Furthermore, let $c_t$ represent the color of the box at time step $t$, and designate $H$ as the maximum number of steps allowable in the game. In the results presented, the baseline methodology confines the context/task space to instances of the form $[a_0^x, a_0^y, b_0^x, b_0^y, c_0 = c_1 = \cdots = c_H]$. The extensive task space contains all possible task configurations denoted as $[a_0^x, a_0^y, b_0^x, b_0^y, c_0, c_1, \dots, c_H]$. The proposed causal curriculum imposes a task space restriction, characterized by tasks of the form $[a_0^x, a_0^y, b_0^x, b_0^y]$, delegating the selection of $[c_1, c_2, \dots, c_H]$ to the environment, predicated upon intrinsic randomness.

Please provide your insights on this aspect.

---

> ### Author Response · Authors · 2023-11-21
>
> Dear reviewer C4uc,
> We thank you for your thoughtful feedback. We will address your comments and questions in detail in the following.
>
> >#### Weaknesses 1
> > “The strategy for avoiding misaligned source tasks, as proposed in this work, relies on causal knowledge about the underlying data-generating mechanisms within the environment. This requirement limits the broader applicability of the strategy, making it dependent on domain-specific knowledge. The practicality of obtaining a causal diagram G for a general target task remains uncertain.”
>
> We appreciate the opportunity to clarifying this issue. First, we note that the knowledge our paper relies on, the causal diagram, is not fundamentally different from the assumption of MDP or POMDP. Formally, when assuming the underlying environment is an MDP or POMDP, a causal diagram is implicitly assumed. For example, if the environment is assumed to be MDP, at each time step there will be three nodes, state $S_t$, action $X_t$ and reward $Y_t$ and we will only have $S_t, X_t$ pointing to $S_{t+1}$.
> Curriculum learning methods already assume access to the domain-specific knowledge of the underlying system dynamics and abilities to manipulate them. Since a causal diagram is well defined from the underlying data-generating mechanisms, one could construct a causal diagram of the environment following the constructive procedure [1, Def. 27.10]. For example, in the Colored Sokoban example, as the curriculum designer, we know that the agent can observe the box color, change the box and itself’s location, each object’s current location, the grid world dynamics and that we might need to push a specific colored box to the location to get rewarded. Given such a level of information, a causal diagram of figure 3 is straightforward to construct. In other words, a causal diagram is not necessarily an additional assumption but an abstraction of the target task dynamics, and thus, a by-product of the assumptions of curriculum learning.
>
> > #### Weakness 2
> > “It remains unclear whether the proposed causal curriculum strategy can be extended to domains with continuous state and action spaces.”
>
> There might be some misreadings on our results and we take this as an opportunity to clarify this point. We first note that the results in the paper (Theorems 1-3) are non-parametric, in the sense that they do not rely on any parametric assumptions of the underlying variable domains and structural causal model. This means that these results are readily applicable to state variables and action variables over continuous domains. To illustrate this point empirically, we conducted extra experiments in a continuous domain to demonstrate the effectiveness of our methods. Please see the common response for the experiment result.

---

> > ### Author Response · Authors · 2023-11-21
> >
> > > #### Question 1
> > > “Re. the Colored Sokoban example:
> > In the baseline, the context/task space is restricted to tasks where the box color remains constant throughout the game. However, considering a more extensive task space that encompasses all possible combinations of box colors might lead the state-of-the-art curriculum strategies to automatically select relevant/aligned tasks during training. In contrast, the proposed causal curriculum limits the task space to initial agent and box positions, with the environment determining box colors based on intrinsic randomness.
> > Formally, let us denote the initial positions of the agent and the box as $(a_0^x, a_0^y)$ and $(b_0^x, b_0^y)$, respectively. Furthermore, let $c_t$ represent the color of the box at time step $t$, and designate $H$ as the maximum number of steps allowable in the game. In the results presented, the baseline methodology confines the context/task space to instances of the form $[a_0^x, a_0^y, b_0^x, b_0^y, c_0=c_1=\cdots=c_H]$. The extensive task space contains all possible task configurations denoted as $[a_0^x, a_0^y, b_0^x, b_0^y, c_0, c_1, \cdots, c_H]$. The proposed causal curriculum imposes a task space restriction, characterized by tasks of the form $[a_0^x, a_0^y, b_0^x, b_0^y]$, delegating the selection of $[c_0, c_1, \cdots, c_H]$ to the environment, predicated upon intrinsic randomness.
> > Please provide your insights on this aspect.”
> >
> > For this more extensive task space that encompasses all possible combinations of box colors, those state-of-the-art curriculum generators will still fail to select causally aligned source tasks. Because in this Colored Sokoban, as long as the box color is intervened by the outside curriculum generator, it will be a misaligned source task (as indicated by our theorem), no matter how the color is intervened. Specifically, in example 1, comparing equation 3 against equation 4 (there was a typo here and we have fixed it in the revised version), we see that in target task, the optimal policy is maximizing the reward distribution of,
> > $$P\left(Y_i = 10 \big| B_i =  \text{next to goal}, C_i, \operatorname{do}\left(X_i = \text{push}\right)\right) = 1[C_i = 0]$$, and
> > $$P\left(Y_i = -10 \big| B_i =  \text{next to goal}, C_i, \operatorname{do}\left(X_i = \text{push}\right)\right) = 1[C_i=1]$$.
> >
> > However, in the color-intervened source task (yellow box), the optimal policy is maximizing the reward distribution of,
> > $$P\left(Y_i = 10 \big| B_i =  \text{next to goal}, \operatorname{do}\left(C_i = 0, X_i = \text{push}\right)\right) = \frac{1}{4}$$, and
> > $$P\left(Y_i = -10 \big| B_i =  \text{next to goal}, \operatorname{do}\left(C_i = 0, X_i = \text{push}\right)\right) = \frac{3}{4}$$.
> > If we intervene the color to be blue, similarly, we have,
> > $$P\left(Y_i = 10 \big| B_i =  \text{next to goal}, \operatorname{do}\left(C_i = 0, X_i = \text{push}\right)\right) = \frac{1}{4}$$, and
> > $$P\left(Y_i = -10 \big| B_i =  \text{next to goal}, \operatorname{do}\left(C_i = 0, X_i = \text{push}\right)\right) = \frac{3}{4}$$.
> > We see that for a curriculum composed of source tasks intervening on the box color, the optimal policy learned is maximizing the reward under the last two distributions instead of the target task distribution. Thus, this more extensive task space also won’t help.
> > Intuitively, because the reward and the box color are confounded by an unobserved random variable (intrinsic randomness), the agent can only access this information by observing the box color. Once the box color is intervened externally, the agent will lose access to this “intrinsic randomness” and cannot know precisely the right time to push the box. In conclusion, even if the baseline methods change the box color at each time step, they still cannot train the agent successfully. From a theoretical perspective, and as characterized by our theory, there are some variables we should not intervene on during the curriculum generation at all; box color in this case falls exactly into this category. We appreciate the opportunity to clarify this sublet and fundamental matter. One of the primary goals of the theoretical framework proposed in this paper is to identify and characterize the variables that cannot be altered without negatively impacting the agent's convergence following training, which constitutes a causally aligned curriculum.
> >
> > [1] Bareinboim, Elias, et al. "On Pearl’s hierarchy and the foundations of causal inference." Probabilistic and causal inference: the works of Judea Pearl. 2022. 507-556.

---

### Author Response · Authors · 2023-11-21
**Common response to all the reviewers**

We appreciate all the valuable suggestions and comments we received, which help us greatly in enhancing the quality of our paper. The revised manuscript has been promptly submitted. We hope that the new version can address your concerns. We also provide a concise overview of the key modifications implemented in the update as follows.

## Experiments in the Continuous Domain

Following your suggestion, we added an experiment in a domain with continuous state/action variables and would like to elaborate on that. First, and as mentioned, we highlight that our augmentation procedure (Theorems 1-3) is non-parametric, which does not restrict the form of state/action variables. Once this procedure is executed, we can continue the pipeline and leverage existing curriculum generators to address more complex domains. Consequently, our proposed method is immediately generalizable to other, more diverse, high-dimensional/continuous environments, as long as they are supported by the curriculum generators being augmented.

Second, the experimental results in the following charts also witness the effectiveness of our augmentation procedure. The experiment environment is a continuous version of the Button Maze (Section 4) where the agent still needs to step onto the goal region at the right time but in a continuous space of states and actions (See Fig. 8 in the appendix for a visualization of the target task). And we report the agent’s target task performance evaluated at different stages of the training by normalized Intermediate Quantile Mean (IQM) with 95% confidence intervals. To compare the benefit of curriculum learning, we also report the agent’s performance trained directly in the target task, which is 0.71$\pm$0.32 in normalized IQM. Then, in the following four charts, we report the agent’s performance trained by each of the curriculum generators before augmentation (original) and after causal augmentation.


|ALP-GMM/Env. Steps       |1K | 50K| 100K | 150K|200K|250K|300K
|-------------------------|---|----|------|-----|----|----|----|
|original                 |0.03$\pm$0.02|0.32$\pm$0.07|0.34$\pm$0.04|0.37$\pm$0.05|0.36$\pm$0.05|0.34$\pm$0.06|0.36$\pm$0.05
|after causal augmentation|0.08$\pm$0.02|0.85$\pm$0.07|0.89$\pm$0.02|0.90$\pm$0.02|0.88$\pm$0.02|0.87$\pm$0.12|0.93$\pm$0.04


|PLR/Env. Steps        |1K | 50K| 100K | 150K|200K|250K|300K
|-------------------------|---|----|------|-----|----|----|----|
|original                 |0.09$\pm$0.07|0.17$\pm$0.15|0.40$\pm$0.03|0.47$\pm$0.16|0.36$\pm$0.02|0.47$\pm$0.09|0.36$\pm$0.21|
|after causal augmentation|0.10$\pm$0.05|0.19$\pm$0.11|0.92$\pm$0.01|0.92$\pm$0.01|0.92$\pm$0.01|0.92$\pm$0.00|0.92$\pm$0.00|

|Goal-GAN/Env. Steps      |1K | 50K| 100K | 150K|200K|250K|300K
|-------------------------|---|----|------|-----|----|----|----|
|original                 |0.07$\pm$0.01|0.29$\pm$0.04|0.33$\pm$0.17|0.53$\pm$0.15|0.55$\pm$0.19|0.59$\pm$0.25|0.53$\pm$0.22|
|after causal augmentation|0.08$\pm$0.04|0.82$\pm$0.03|0.88$\pm$0.02|0.88$\pm$0.01|0.87$\pm$0.02|0.86$\pm$0.02|0.86$\pm$0.11|


|Currot/Env. Steps           |1K | 50K| 100K | 150K|200K|250K|300K
|-------------------------|---|----|------|-----|----|----|----|
|original                 |0.08$\pm$0.02|0.44$\pm$0.24|0.64$\pm$0.23|0.61$\pm$0.19|0.63$\pm$0.20|0.70$\pm$0.23|0.64$\pm$0.19|
|after causal augmentation| 0.07$\pm$0.03|0.88$\pm$0.08|0.89$\pm$0.03|0.95$\pm$0.06|0.96$\pm$0.07|0.97$\pm$0.00|0.96$\pm$0.08|

We can see that agents trained by vanilla curriculum generators (original) failed to learn the optimal target policy and thus are outperformed by those trained directly in the target task. This indicates that training with misaligned source tasks is hurting the agent’s performance. Nonetheless, the agents trained by the causally-augmented (labeled as causal) curriculum generators all converged to the optimal, even surpassing the one trained directly in the target task. Additionally, after augmentation, the narrower confidence interval of the agent’s performance at convergence suggests that agents trained in this way are also more stable than both training directly in the target and training with non-causal curriculum generators. This experiment empirically corroborates with our theoretical finding, namely, that the proposed augmentation procedure is applicable to continuous domains. For more details, please refer to section F in the appendix and Fig. 9 for a nicer graphical view of the experiment results.

---

> ### Author Response · Authors · 2023-11-21
>
> ## Other Changes in the Revised Version
> 1. Improved figure captions. We added more details to make sure that each figure is easier to understand on their own.
> 2. Related works. We added more related works to section G in the appendix as suggested by some reviewers. We also added a pointer in the introduction section leading to this extended section on related works.
> 3. Clarity of the paper.
> - In the preliminary section, we improved the sentence explaining the graph theory notations about $ch,\ de,\ an$.
> - In section 2, we removed “*” from Def. 1 SCM $M$.
> - In section 2, we highlighted the difference between figure 3(a) and 3(b) by coloring the edit operators.
> - In section 2 example 1, we changed the word “fixing” to “intervening” to reduce confusion.
> - In the appendix section A, we added a flow chart for an overview of our augmentation procedure.
> - In the appendix section H, we discussed the relationship between different environment modeling methods: MDPs, POMDPs and SCMs.
> - In the appendix section I, we added a brief explanation for the limitations of our method and potential extensions in the future.
> 4. More experiment results. In the appendix section F, we added the results and implementation details of the continuous Button Maze.

---

### Meta-Review · Area_Chair_D3b3 · 2023-12-06

**Metareview:**

The submitted paper proposes an approach for curriculum design for training RL agents based on a causal perspective. To this end, the paper proposes notions of aligned tasks, valid edits, and causally aligned curricula, that ensure that the optimal decisions for individual states are "consistent" and direct the learning agent in the right direction.
The ratings for this paper are quite mixed (reject, accept, marginally below acceptance, marginally above acceptance) but not all reviewers reacted to the authors' rebuttal which, in my opinion, addressed the critical reviewers' main concerns sufficiently.

The main strength of the submitted paper is certainly its methodological contribution, bringing a well-motivated approach for curriculum alignment into the field which differs conceptually from existing approaches. The authors also provide proof-of-concept experiments illustrating the performance of their approach against several baselines on simple environments with latent confounding.

On the negative side, the paper is partly not well presented. Concretely, there is a heavy notation that is needed to derive the theoretical statements (and mainly stems from the causality) but little effort was made to bridge the gap with RL notation which led to several confusions during review and made the paper partly hard to read. Furthermore, while the authors made some effort to illustrate their concepts and ideas through figures, they need to carefully rework figure captions and make sure to pick as little confusing examples as possible to enable the reader to grasp the ideas. Last but not least, the experiments are sufficient for a proof-of-concept but don't shed a lot of detail into details of the proposed method regarding scalability but also robustness, e.g., if the causal graph is not precisely known.

Deciding on this paper was not easy but I think that the methodological approach (and its differences to existing work) warrant acceptance of the paper. Nevertheless, the authors are strongly encouraged to carefully update their paper in line with the above comments and the other reviewers's remarks.

**Justification For Why Not Higher Score:**

Certain aspects could be understood deeper, e.g. regarding scalability or incorrect causal graphs.

**Justification For Why Not Lower Score:**

Decent paper that provides a new perspective on the curriculum generation problem.

---

### Decision · Program_Chairs · 2024-01-16

Accept (poster)